# Stochastic Approximation Algorithms for Systems of Interacting Particles

**Mohammad Reza Karimi**
ETH Zürich
mkarimi@inf.ethz.ch

**Ya-Ping Hsieh**
ETH Zürich
yaping.hsieh@inf.ethz.ch

**Andreas Krause**
ETH Zürich
krausea@ethz.ch

## Abstract

Interacting particle systems have proven highly successful in various machine learning tasks, including approximate Bayesian inference and neural network optimization. However, the analysis of these systems often relies on the simplifying assumption of the *mean-field* limit, where particle numbers approach infinity and infinitesimal step sizes are used. In practice, discrete time steps, finite particle numbers, and complex integration schemes are employed, creating a theoretical gap between continuous-time and discrete-time processes. In this paper, we present a novel framework that establishes a precise connection between these discrete-time schemes and their corresponding mean-field limits in terms of convergence properties and asymptotic behavior. By adopting a dynamical system perspective, our framework seamlessly integrates various numerical schemes that are typically analyzed independently. For example, our framework provides a unified treatment of optimizing an infinite-width two-layer neural network and sampling via Stein Variational Gradient descent, which were previously studied in isolation.

## 1 Introduction

Dynamics of interacting particle systems are central to many challenges in modern machine learning. These range from algorithm design for approximate Bayesian inference, to the study of equilibria in games [26, 31, 34]. Moreover, researchers have gained valuable insights by interpreting the training process of over-parametrized two-layer neural networks as a system of interacting particles, thereby advancing our intuition in this domain [11, 12, 36–38, 46, 51].

Analyzing the behavior of such systems poses a significant challenge. To simplify the analysis, researchers often turn to the concept of the *mean-field limit*. In this approach, the number of particles is increased to infinity, the step-size of the algorithm shrinks to zero, and results are deduced from this continuous limit. Specifically, an initial probability distribution is assigned to the infinitely many particles, and a continuous-time evolution is derived for this distribution. The resulting evolutionary equation serves as a powerful tool for gaining insights into the system's asymptotic behavior, significantly aiding in the comprehension of systems with a large number of interacting particles [1, 5, 22, 26, 47].

Although the mean-field limit has offered valuable insights into the aforementioned problems, its rigorous justification necessitates examining systems with both a *finite* number of particles and the *discrete-time algorithms* commonly used in practice. Unfortunately, to the best of our knowledge, while a substantial body of work exists on finite-particle systems (see, e.g., the review papers [8, 9] and references therein), these studies are confined to analyzing continuous-time dynamics. As a result, a gap between theory and practice has emerged, and a precise link between continuous and discrete-time schemes is still largely lacking.

**§ Contributions.** Our paper aims to bridge this gap by rigorously establishing the convergence of discrete-time algorithms to their continuous-time counterparts in terms of long-term behavior.

37th Conference on Neural Information Processing Systems (NeurIPS 2023).

To accomplish this, we draw inspiration from the field of *dynamical system* theory, which traces its origins back to early developments in statistics [42] and has recently found success in various domains of machine learning, including optimization theory, games, and sampling [21, 23, 24].

This dynamical system framework offers two significant advantages. Firstly, it provides a flexible framework capable of accommodating a wide range of practical schemes commonly employed in real-world scenarios. Secondly, this framework enables a unified treatment of various machine learning tasks that were previously studied in isolation. Notably, it allows us to address tasks such as training neural networks and approximate Bayesian inference using techniques like Stein variational gradient descent in a coherent and unified manner.

In summary, our paper makes the following contributions:

1. We introduce a comprehensive framework for analyzing a broad class of algorithms called *stochastic approximation schemes*. These schemes are widely utilized in simulating systems of interacting particles, and our framework provides a unified approach to analyze their behavior.

2. Under mild assumptions on the discrete-time schemes and the mean-field dynamics, we prove the convergence of these schemes towards their respective mean-field limits. The convergence is established in terms of the *2-Wasserstein distance*, a metric commonly used to measure the dissimilarity between probability distributions.

3. Since our framework is specifically tailored to address a wide range of fields, we instantiate our main theorem and provide novel guarantees to a diverse array of interacting particle systems across domains such as the Stein variational gradient descent, the training of wide two-layer neural networks, and the examination of game equilibria.

## 2 Illustration: Training Two-Layer Neural Networks via Noisy SGD

To illustrate the particle system under study and its corresponding mean-field limit, let us consider the example of training a two-layer neural network with $N$ neurons and squared loss using the noisy SGD algorithm. The network takes an input $z \in \mathbb{R}^{d-1}$ and computes function:

$$h_{\boldsymbol{\theta}}(z) := \frac{1}{N} \sum_{i=1}^{N} \varphi(\theta^i, z), \quad \text{with} \quad \theta^i = (a^i, b^i) \in \mathbb{R}^{d-1} \times \mathbb{R} \quad \text{and} \quad \varphi(\theta^i, z) = b^i \kappa(\langle a^i, z \rangle).$$

Here, $\boldsymbol{\theta} = (\theta^1, \dots, \theta^N)$ denotes the collection of neurons' parameters, and $\kappa$ is an activation function (such as sigmoid or tanh). To train this network, one ideally minimizes the regularized risk $L(\boldsymbol{\theta}) := \mathbb{E}_{(y,z)\sim\mathcal{D}} \frac{1}{2}(y - h_{\boldsymbol{\theta}}(z))^2 + \lambda U(\boldsymbol{\theta})$, where $\mathcal{D}$ is the data distribution, and $U(\boldsymbol{\theta}) = \frac{1}{N} \sum_i U(\theta^i)$ is a regularizer, such as $U(\theta) = \frac{1}{2}|\theta|^2$. Applying the noisy gradient descent algorithm to the regularized risk then results in the update rule:

$$\theta_{k+1}^i = \theta_k^i - \gamma_{k+1} \nabla_{\theta^i} L(\boldsymbol{\theta}_k) + \sigma\sqrt{2\gamma_{k+1}}\, \xi_{k+1}^i, \qquad i = 1, \dots, N, \tag{SGD$_\sigma$}$$

where $\gamma_{k+1}$ is the step-size, $\sigma \geq 0$ is the noise level, and $\{\xi_{k+1}^i\}$ is a collection of i.i.d. standard Gaussians. To cast this algorithm as a system of interacting particles, we define the *interaction kernel* $W$ and the *potential* $V$ as

$$W(\theta, \theta') = \mathbb{E}_{z\sim\mathcal{D}}[\varphi(\theta, z)\varphi(\theta', z)] \quad \text{and} \quad V(\theta) = \mathbb{E}_{(y,z)\sim\mathcal{D}}[y\,\varphi(\theta, z)] + \lambda U(\theta). \tag{1}$$

It is then easily seen (cf. [36, Eqn. 4]) that the update rule (SGD$_\sigma$) can be rewritten as

$$\theta_{k+1}^i = \theta_k^i - \gamma_{k+1} \left( \frac{1}{N} \sum_{j=1}^{N} \nabla_\theta W(\theta_k^i, \theta_k^j) + \nabla V(\theta_k^i) \right) + \sigma\sqrt{2\gamma_{k+1}}\, \xi_{k+1}^i, \quad i = 1, \dots, N. \tag{2}$$

By considering neurons as particles, the equation (2) reveals that the training dynamics of each particle is influenced by the potential $V$ and the interaction energy $W$ with other particles. This interpretation highlights that (SGD$_\sigma$) represents the evolution of a system of interacting particles. In a broader context, one may consider more sophisticated algorithms, several of which (see Section 4 for examples) can be formulated as

$$\theta_{k+1}^i = \theta_k^i - \gamma_{k+1} \left( \nabla_{\theta^i} L(\boldsymbol{\theta}_k) + P_{k+1}^i \right) + \sigma\sqrt{2\gamma_{k+1}}\, \xi_{k+1}^i, \qquad i = 1, \dots, N \tag{3}$$

where $P_{k+1}^i$ is the (random or deterministic) perturbation in evaluating the gradient. These algorithms are usually called *stochastic approximation algorithms*. For example, Noisy SGD corresponds to setting $P_{k+1}^i = \nabla_{\theta^i} \widetilde{L}(\boldsymbol{\theta}_k) - \nabla_{\theta^i} L(\boldsymbol{\theta}_k)$, where $\widetilde{L}$ is the loss of a random batch of data.

**§ Continuous-time limit and the mean-field approximation.**   We now go from the discrete-time algorithm (3) to a continuous-time process and then derive the mean-field approximation. First, observe that the risk $L(\boldsymbol{\theta})$ depends on $(\theta^1, \ldots, \theta^N)$ only through their *empirical measure* $\widehat{\mu} = \frac{1}{N} \sum_i \delta_{\theta^i}$. For example,

$$\frac{1}{N} \sum_{j=1}^{N} \nabla_\theta W(\theta, \theta^j) + \nabla V(\theta) = \int \nabla_\theta W(\theta, \theta') \, \widehat{\mu}(d\theta') + \nabla V(\theta) =: b(\theta, \widehat{\mu}), \tag{4}$$

where we have introduced the *drift* function $b$. To simplify the analysis, previous studies then consider the idealized setting where an infinitesimal step-size is employed, thereby further reducing (3) to the following system of stochastic differential equations (SDEs):

$$d\theta_t^i = b(\theta_t^i, \widehat{\mu}_t) \, dt + \sigma \sqrt{2} \, dW_t^i, \qquad i = 1, \ldots, N, \tag{5}$$

where $\widehat{\mu}_t = \frac{1}{N} \sum_i \delta_{\theta_t^i}$ is the empirical measure of the particles at time $t$, and $\{W_\cdot^i\}$ is a collection of i.i.d. standard Brownian motions. In the over-parametrized regime where the number of particles $N$ becomes very large, one can approximate the initial setting of the particles $\widehat{\mu}_0$ with a probability density $\rho_0$, and consider the following *mean-field* approximation defined over $\mathbb{R}^d$ as

$$d\theta_t = b(\theta_t, \rho_t) \, dt + \sigma \sqrt{2} \, dW_t, \quad \rho_t = \text{density}(\theta_t). \tag{6}$$

This dynamics captures the behavior of an individual particle $\theta$ within a density of particles distributed according to $\rho$. The analysis of (6) turns out to be considerably simpler compared to the system of SDEs in (5), e.g., via studying the evolutionary PDE $\partial_t \rho_t = \nabla \cdot (\rho_t \, b(\cdot, \rho_t)) + \sigma^2 \Delta \rho_t$. As a result, it has attracted significant interest in the field of deep learning theory [11, 12, 36–38].

In conclusion, the mean-field dynamics (6) offers a powerful and elegant framework, but its validity rests on two simplifying assumptions: an infinite number of particles and a step-size approaching zero. A rigorous justification of these two steps is by no means trivial. While the existing literature has made progress in addressing the infinite particle issue, the second assumption has received comparatively less attention. Bridging this gap is one of the primary objectives of our paper. Specifically, we aim to establish the Wasserstein convergence of the discrete-time dynamics (3) to the same limit sets[1] as the continuous-time particle dynamics (5), under mild conditions on the drift $b$ and perturbations $\{P_{k+1}^i\}$, as well as the step-size rule $\gamma_{k+1}$.

## 3   Dynamics of Systems of Interacting Particles

This section presents the fundamental master theorem that forms the basis for all the applications discussed in Section 4. Our objectives are two-fold. Firstly, we aim to provide a set of assumptions for the discrete-time scheme that can be easily verified by practical algorithms. Secondly, we establish a set of assumptions on the *mean-field* dynamics, which, as demonstrated in Section 4, are readily implied by the standard assumptions in the *finite particle* regime. The key result of our paper asserts that, under these assumptions, any discrete-time scheme converges to its continuous counterpart, thus closing the existing theoretical gap.

### 3.1   The algorithmic template

In this paper, we study the *stochastic approximation algorithms* for simulating systems of interacting particles of the form:

$$x_{k+1}^i = x_k^i + \gamma_{k+1} \big\{ b(x_k^i, \widehat{\mu}_k) + P_{k+1}^i \big\} + \sqrt{\gamma_{k+1}} \sigma(x_k^i, \widehat{\mu}_k) \, \xi_{k+1}^i, \qquad i = 1, \ldots, N, \tag{SAA}$$

where $b : \mathbb{R}^d \times \mathscr{P}_2(\mathbb{R}^d) \to \mathbb{R}^d$ is the (non-local) drift,[2] $\sigma : \mathbb{R}^d \times \mathscr{P}_2(\mathbb{R}^d) \to \mathbb{R}^{d \times d}$ is the (state-dependent and non-local) diffusion coefficient, $P_{k+1}^i$ is the noise and bias in evaluating the drift, and $\widehat{\mu}_k$ is the empirical measure of the particles at iteration $k$. The system (SAA) can be written in a more succinct way by stacking all the variables in a larger vector. That is, define $x := (x^i)_{i \in [N]} \in (\mathbb{R}^d)^{\otimes N}$, and let $\mu_x$ be the empirical distribution corresponding to the $N$ vectors in $x$, and define the *aggregated drift and diffusion terms* as

$$\boldsymbol{b}(x) := (b(x^i, \mu_x))_{i \in [N]} \in (\mathbb{R}^d)^{\otimes N}, \quad \boldsymbol{\sigma}(x) := \text{diag}((\sigma(x^i, \mu_x))_{i \in [N]}). \tag{7}$$

Define $\{P_{k+1}\}$ and $\{\xi_{k+1}\}$ analogously. We can then rewrite (SAA) as:

$$x_{k+1} = x_k + \gamma_{k+1} \{ \boldsymbol{b}(x_k) + P_{k+1} \} + \sqrt{\gamma_{k+1}} \, \boldsymbol{\sigma}(x_k) \, \xi_{k+1}. \tag{PSAA}$$

---

[1]The *limit set* of a curve $(c(t))_{t \geq 0}$ in a metric space is $\bigcap_{t \geq 0} \text{cl}(c([t, \infty)))$, that is, the set of all limits of convergent sequences $\{c(t_k), t_k \to \infty\}$.

[2]$\mathscr{P}_2(\mathbb{R}^d)$ is the space of probability measures on $\mathbb{R}^d$ with bounded second moments.

## 3.2 Dynamical system theory

By considering infinitesimal step-sizes and neglecting the perturbations $P^i_{k+1}$, we can derive a system of corresponding SDEs as follows:

$$dX^i_t = b(X^i_t, \widehat{\mu}_t)\, dt + \sigma(X^i_t, \widehat{\mu}_t)\, dW^i_t, \quad i = 1, \ldots, N, \quad \widehat{\mu}_t = \tfrac{1}{N}\sum_i \delta_{X^i_t}, \qquad \text{(Sys-SDE)}$$

with the corresponding aggregated version

$$dX_t = \boldsymbol{b}(X_t)\, dt + \boldsymbol{\sigma}(X_t)\, dW_t. \qquad \text{(PSDE)}$$

It is important to emphasize that even though we have rearranged the particles into a unified vector and introduced the concepts of aggregated drift and diffusion, the original process retains its *exchangeability*. This property implies that the distribution of $X^i_t$ in (Sys-SDE) remains invariant under permutations of the particles (for further details, refer to [8, Def. 2.1]).

The primary objective of our paper is to rigorously establish the convergence of the stochastic approximation scheme (PSAA) to its continuous-time counterpart (PSDE). To accomplish this, we employ the *dynamical system* theory introduced by Benaïm and Hirsch [4]. First, we construct a continuous-time *interpolated process* associated with the discrete-time algorithm (PSAA):

$$X_t = x_k + (t - \tau_k)(\boldsymbol{b}(x_k) + \mathbb{E}[P_{k+1} \mid \mathscr{F}_t]) + \boldsymbol{\sigma}(x_k)(W_t - W_{\tau_k}), \quad \tau_k \le t < \tau_{k+1}. \qquad \text{(Int)}$$

Here, $\tau_k = \sum_{j=1}^k \gamma_j$ represents the cumulative time until step $k$. It is worth noting that we construct (Int) in such a way that it is adapted to the same filtration $(\mathscr{F}_t)$ as the Brownian motion.

To compare (PSAA) with (PSDE), we integrate (PSDE) using the following approach: For a fixed $t \ge 0$, we define $W^{(t)}_s = W_{t+s} - W_t$ and denote the solution of (PSDE) as the *flow*:

$$\Phi^{(t)}_s = X_t + \int_0^s \boldsymbol{b}(\Phi^{(t)}_u)\, du + \int_0^s \boldsymbol{\sigma}(\Phi^{(t)}_u)\, dW^{(t)}_u. \qquad \text{(Flow)}$$

It is important to observe that the flow *starts at $X_t$* and continues according to the true SDE.

We now introduce the central concept in our paper, which is the *asymptotic pseudotrajectory* theory of Benaïm and Hirsch [4].

**Definition 1** (Wasserstein asymptotic pseudotrajectory). We say the stochastic process $(X_t)_{t \ge 0}$ is a *Wasserstein asymptotic pseudotrajectory* (WAPT) of the flow $\Phi$ if for any fixed $T > 0$,

$$\lim_{t \to \infty} \sup_{0 \le s \le T} \mathcal{W}_2(X_{t+s}, \Phi^{(t)}_s) = 0, \qquad (8)$$

where $\mathcal{W}_2(\cdot, \cdot)$ denotes the 2-Wasserstein distance between two distributions.

The notion of WAPT provides a measure of "asymptotic closeness" between two stochastic processes. In particular, (8) requires that $(X_t)_{t \ge 0}$ closely tracks the flow $\Phi^{(t)}_s$ over arbitrarily long time intervals $T$ with arbitrary precision. The key aspect of the WAPT is that it serves as a tool specifically designed to establish the convergence of a stochastic approximation scheme to its continuous-time counterparts. In particular, it is known that proving the convergence of a stochastic approximation algorithm (PSAA) to its continuous-time counterparts (PSDE) can be accomplished by demonstrating the following two conditions [3, 4]:

- The interpolation (Int) satisfies the WAPT condition with respect to the corresponding flow.

- The iterates $\{x_k\}_k$'s in (PSAA) have bounded second moments.

Below, we present a set of general conditions that are straightforward to verify and ensure the satisfaction of the above two conditions.

*Remark.* There are two major reasons for choosing $\mathcal{W}_2$ as the distance in (8): Firstly, the 2-Wasserstein space is a *metric space* on which McKean–Vlasov equations can be seen as a *flow*, both aspects indispensable for invoking the dynamical system theory of Benaïm and Hirsch [3, 4]. Secondly, it is a popular metric in the propagation of chaos literature. This allows a seamless transition from convergence guarantees for stochastic approximation schemes to their mean-field limit counterparts via combining our results with the propagation of chaos results in the literature, see (13).

### 3.3 Technical Assumptions

We proceed to present our technical assumptions and discuss their generality.

**§ On the mean-field dynamics.** We begin by introducing three assumptions that pertain to the drift and diffusion coefficients of the continuous-time dynamics (Sys-SDE):

**Assumption 1** (Lipschitzness of drift and diffusion). *There is some $L > 0$ such that for all $x, y \in \mathbb{R}^d$ and all $\mu, \nu \in \mathscr{P}_2(\mathbb{R}^d)$,*

$$|b(x, \mu) - b(y, \nu)| + \|\sigma(x, \mu) - \sigma(y, \nu)\|_F \leq L(|x - y| + \mathcal{W}_2(\mu, \nu)).$$

**Assumption 2** (Drift growth condition). *For all $\mu \in \mathscr{P}_2(\mathbb{R}^d)$, there is some $C_v > 0$ such that*

$$\int \langle x, b(x, \mu) \rangle \, \mu(dx) \leq C_v \int (|x| + 1) \, \mu(dx).$$

**Assumption 3** (Boundedness of the diffusion). *There is some $K > 0$ such that for all $x \in \mathbb{R}^d$ and all $\mu \in \mathscr{P}_2(\mathbb{R}^d)$, $\|\sigma(x, \mu)\|_F \leq K$.*

We note that the first assumption is standard and is commonly used to prove existence of strong solutions for the mean-field equation (see [26, Thm. 3.3]). The other two assumptions are exceedingly weak and are satisfied by all the applications we consider.

**§ On the stochastic approximation schemes.** The following assumptions concern the time-discretization scheme and the induced noise and bias.

**Assumption 4** (Noise and bias). *The perturbation $P_{k+1}$ decomposes into noise and bias as $P_{k+1} = U_{k+1} + \varepsilon_{k+1}$, where the noises $\{U_{k+1}\}$ form a martingale difference sequence, i.e., $\mathbb{E}[U_{k+1} \mid U_k] = 0$, and have second moments uniformly bounded by $M_U$. In addition, the bias terms satisfy $\varepsilon_k \in \mathscr{F}_{\tau_k}$ and*

$$\mathbb{E}[|\varepsilon_{k+1}|^2 \mid \mathscr{F}_{\tau_k}] = \mathcal{O}(\gamma_{k+1}^2 |\boldsymbol{b}(x_k)|^2 + \gamma_{k+1}). \tag{9}$$

**Assumption 5** (Step-sizes). *The step-sizes are decreasing and satisfy the Robbins-Monro summability conditions*

$$\sum_k \gamma_{k+1} = \infty \quad \text{and} \quad \sum_k \gamma_{k+1}^2 < \infty. \tag{10}$$

*Moreover, we require, for some constant $P > 0$,*

$$\gamma_{k+1}/\gamma_k + P\gamma_k\gamma_{k+1} \leq 1 - \gamma_{k+1}. \tag{11}$$

The condition specified in equation (9) is algorithm-dependent, and as we will prove in Section 4, it is satisfied by numerous practical schemes. In Assumption 5, equation (10) is a commonly used condition in the literature [42], while (11) imposes a mild growth condition on the step size, which remains satisfied even for slowly-decreasing step-sizes such as $\gamma_{k+1} \sim (\sqrt{k} \log k)^{-1}$. Therefore, this condition is not overly restrictive and accommodates a wide range of scenarios.

**§ On dissipativity.** In the context of dynamical system theory, it is important to ensure that the iterates of stochastic approximation schemes have bounded second moments. In the literature, this requirement is often met by imposing *dissipativity*-type conditions. Building upon this concept, we introduce the following definition:

**Definition 2** (Average Dissipativity). *We call the drift $b$ to be $(\alpha, \beta)$-dissipative on average for some $\alpha > 0$ and $\beta \in \mathbb{R}$, if for all probability measures $\mu \in \mathscr{P}_2(\mathbb{R}^d)$, it holds*

$$\int \langle x, b(x, \mu) \rangle \, \mu(dx) \leq -\alpha \int |x|^2 \, \mu(dx) + \beta.$$

The concept of average dissipativity, as introduced in Definition 2, provides a novel formulation specifically designed to capture the dissipativity property of a drift function that depends on *measures*, as in equation (Sys-SDE). In contrast to the traditional notion of dissipativity [19], which focuses on the dissipative behavior of *individual* particles in isolation, this formulation allows for a more fine-grained control over the *collective* behavior exhibited by $N$ particles, each running in parallel with the same stochastic approximation scheme. In Section 4, we will provide concrete examples from the applications of machine learning to demonstrate the satisfaction of average dissipativity.

## 3.4 Main Results

We are now ready to state our main theorem, whose proof can be found in Appendix B.

**Theorem 1.** *Consider the algorithm* (SAA)*, where the drift b and diffusion $\sigma$ satisfy Assumptions 1–3, and the step-sizes $\{\gamma_{k+1}\}$ and the perturbations $\{P^i_{k+1}\}$ satisfy Assumptions 4 and 5. Then the following holds:*

- *The interpolation* (Int) *of iterates of the algorithm is a WAPT of the flow in* (Flow).

- *Moreover, if the drift b is dissipative on average (see Definition 2), the iterates of* (PSAA) *are bounded in second moments, and their limit set is included in that of the original SDE* (PSDE).

*Proof sketch.* The main step of the proof is the construction of the *Picard process*, which is inspired by [23] and defined as follows:

$$\Pi^{(t)}_s = X_t + \int_0^s b(X_{t+u})\, du + \int_0^s \sigma(X_{t+u})\, dW^{(t)}_u. \tag{Picard}$$

The proof is completed in four steps: first, we prove that the Picard process closely tracks the flow (Flow), and then we bound the distance between the Picard process and the interpolation (Int). By using martingale convergence arguments, employing our assumption on bias Assumption 4, and Grönwall inequality, we conclude the proof of the WAPT property. Lastly, we show that dissipativity on average ensures a uniform bound on the second moments of the iterates. The convergence is then implied by invoking [23, Theorem 3]. ∎

In summary, according to Theorem 1, the convergence of the stochastic approximation scheme in (SAA) can be reduced to its continuous-time counterpart in (Sys-SDE) when the assumptions stated in Section 3.3 are satisfied.

Our Theorem 1 can be combined with existing results in the finite particle regime to establish the overall convergence towards the desired mean-field limit. To see this, let $M^{i,N}_t$ be $N$ independent processes, synchronously coupled with (Sys-SDE), each starting from $X^i_0$ and following

$$dM^{i,N}_t = b(M^{i,N}_t, \mu_t)\, dt + \sigma(M^{i,N}_t, \mu_t)\, dW^{i,N}_t,$$

where $\mu_t$ is the *mean-field solution*. A common phenomenon in the study of interacting particle systems, known as *uniform propagation of chaos* [8, 22, 27, 47], implies the existence of a constant $C$ such that for every $N$:

$$\sup_{t\geq 0} \frac{1}{N} \sum_{i=1}^N \mathcal{W}_2^2(X^{i,N}_t, M^{i,N}_t) \leq \frac{C}{N}, \tag{12}$$

where $X^{i,N}_t$ represents the particles following the continuous-time dynamics in (Sys-SDE). Letting $\mu_\infty$ represent the limit of the mean-field equation, a straightforward application of the triangle inequality argument, which we defer to Appendix B.2, yields:

$$\lim_{k\to\infty} \frac{1}{N} \sum_{i=1}^N \mathcal{W}_2^2(x^i_k, \mu_\infty) \leq \frac{C}{N} \to 0 \quad \text{as} \quad N \to \infty. \tag{13}$$

In other words, as the number of particles approaches infinity, the law of the empirical distribution of the particles following the discrete-time algorithm (SAA) also converges to the mean-field solution.

**§ A note on the literature.** The rich body of literature on McKean-Vlasov SDEs and interacting particle systems offers considerable insights on the convergence of Euler-Maruyama and Milstein type numerical schemes to their limiting mean-field equations, e.g., [2, 28, 41]. However, it is noteworthy that our study diverges in several key respects: Firstly, our work emphasizes on generic *stochastic* and *biased* drift oracles. This contrasts with the deterministic and unbiased drift oracles considered in the aforementioned studies, making our algorithmic approach broader in scope. Secondly, while those studies present strong finite-time error bounds, our convergence results focus on providing asymptotic guarantees. Lastly, we incorporate different underlying assumptions. For instance, we need global Lipschitz drifts to ensure globally integrable flows, while one-sided Lipschitz drifts are allowed in the works by [2, 28, 41]. However, our growth condition in Assumption 2 requires control on average, whereas the works mentioned assume stronger pointwise controls.

In the light of these distinctions, we believe that our work complements this body of literature.

# 4 Applications

The goal of this section is to demonstrate the wide-ranging applicability of our framework across diverse domains such as machine learning, game theory, and physics, which were previously analyzed in isolation. In each of these applications, we demonstrate that standard assumptions in their respective domains meet the necessary conditions to invoke Theorem 1. All proofs are deferred to Appendix C.

## 4.1 Two-Layer Neural Networks and Mean-Field Langevin

Our first application is providing a rigorous guarantee for the training dynamics of wide two-layer neural networks as alluded to in Section 2. To begin, let us quickly recall the notations therein: The noisy SGD iterates in ($SGD_\sigma$) for wide two-layer neural networks can be viewed as approximations of the mean-field dynamics (6) through the discrete-time system (SAA), whose drift term $b(\cdot, \cdot)$ is defined in (1). The corresponding continuous-time and finite-particle dynamics is (5), which has been extensively studied in recent years; see [11, 12, 36–38, 46, 51] and references therein. Under the standard assumptions as in [11, 12, 37], a simple application of Theorem 1 then yields the convergence of ($SGD_\sigma$):

**Corollary 1.** *Let $\kappa(\cdot)$ denote the activation function. Assume that (1) $\kappa$ and $\kappa'$ are Lipschitz and bounded, (2) the data has bounded support, and (3) $|a \kappa'(a)|$ is bounded. Then the discrete-time scheme ($SGD_\sigma$) converges in $\mathcal{W}_2$ to the same limit sets as the continuous-time (5).*

The mean-field dynamics for two-layer neural networks converges to a stationary point in Wasserstein space due to its gradient flow structure [12]. In addition, it is known that under fairly mild assumptions, the resulting limit becomes a unique *global* risk minimizer [12]. By employing uniform propagation of chaos for neural networks [46], it is observed that the limit sets of the continuous-time $N$-particle dynamics exhibit similar generalization error to the global minimizer. Remarkably, Corollary 1 validates that noisy SGD and other discretizations following template (SAA) eventually converge to this limit set, providing justification for the observed generalization behavior in neural networks.

**§ Comparison to prior work.** Our result above cannot be directly compared to the existing analysis conducted on the discrete-time scheme ($SGD_\sigma$), namely [37], due to the assumption A1 therein, which excludes step-size rules of the form $\gamma_{k+1} \sim k^{-\beta}$ where $\beta \in (1/2, 1]$. Conversely, the bounds provided in [37] for a *constant* step-size are non-asymptotic (albeit doubly-exponential), resulting in stronger conclusions compared to our asymptotic results. Therefore, these two analyses complement each other and contribute to a more comprehensive understanding of the training dynamics exhibited by neural networks.

## 4.2 Stein Variational Gradient Descent

Sampling from a distribution $\pi \propto e^{-V}$ is a crucial task in various machine learning applications. An effective method that has demonstrated practical success in this regard is the *Stein variational gradient descent* (SVGD) [29, 30]. Intuitively, this method emulates the steepest descent for the KL divergence in *continuous-time* with a *continuous probability measure*. In practice, the algorithm is implemented as an interacting particle system as follows (see [33] for a derivation). Let $K : \mathbb{R}^d \times \mathbb{R}^d \to \mathbb{R}$ be a *positive definite kernel*. The SVGD algorithm updates the set of $N$ particles as

$$x_{k+1}^i = x_k^i - \frac{\gamma_{k+1}}{N} \sum_{j=1}^N \left( K(x_k^i, x_k^j) \nabla V(x_k^j) - \nabla_2 K(x_k^i, x_k^j) \right), \qquad i = 1, \dots, N, \qquad (SVGD_k)$$

where $\nabla_2 K$ is the gradient of $K$ with respect to its second input. The corresponding continuous-time dynamics is then:

$$\frac{d}{dt} X_t^i = \frac{1}{N} \sum_j \nabla_2 K(X_t^i, X_t^j) - \frac{1}{N} \sum_j K(X_t^i, X_t^j) \nabla V(X_t^j), \qquad i = 1, \dots, N, \qquad (SVGD_t)$$

which is a special case of (Sys-SDE) with $b(x, \mu) = (\nabla_2 K * \mu)(x) - (K * (\mu \nabla V))(x)$ and $\sigma \equiv 0$ (no diffusion), where $*$ denotes the convolution operator. We now prove:

**Corollary 2.** *Suppose that (1) $\nabla V(x)$ is Lipschitz, (2) $V$ is dissipative,[3] (3) for some $C > 0$, $|\nabla V(x)| \leq C(1 + |x|^2)$, (4) $\|K\|_\infty, \|\nabla_2 K\|_\infty, \|\nabla^2 K\|_\infty < \infty$, (5) $|\nabla_2 K(x, y)| \leq \eta/|x - y|$, and (6) $K(x, y) \leq \eta/|x - y|^2$ for some $\eta > 0$. Then the iterates in ($SVGD_k$) converge in $\mathcal{W}_2$ to the same limit sets as the continuous-time process ($SVGD_t$).*

---

[3] That is, $\exists m > 0$, $m' \in \mathbb{R}$ such that $\langle x, \nabla V(x) \rangle \geq m|x|^2 - m'$.

Similar to neural networks training discussed in Section 4.1, the mean-field SVGD also exhibits a gradient flow structure, and in this case, the target distribution $\pi$ serves as the only limit set for the mean-field SVGD [29]. Combining Corollary 2 with a uniform propagation of chaos argument for SVGD then confirms that the iterates of (SVGD$_k$), or any discretization of (SVGD$_t$) that satisfies Assumption 4, effectively converge to a distribution that closely approximates $\pi$ in terms of $\mathcal{W}_2$. This outcome underscores the effectiveness of these algorithms in sampling.

§ **Comparison to prior work.** While there has been extensive theoretical work on the convergence of SVGD, most of it focuses on either the *population limit* (i.e., when $N \to \infty$ in (SVGD$_k$)) or the *vanishing step-size* (which directly examines the properties of (SVGD$_t$)) [10, 14, 25, 29, 43, 45]. However, the convergence behavior of the discrete iterates (SVGD$_k$) remains a challenging task with limited success. To the best of our knowledge, the only existing work in this direction is [44], but its result is not directly comparable to ours. Although our assumptions on the bounded derivatives of $V$ and $W$ are similar to those in [44], the difference lies in the requirement imposed on the target distribution $e^{-V}$. Specifically, Shi and Mackey [44] assume a T1-inequality on $e^{-V}$ [50], whereas our work requires $V$ to be dissipative. Notably, our approach holds a significant advantage over [44] in terms of simplicity: The result of [44] is only applicable to a highly specific and fairly complicated step-size rule (see [44, Cor 2]), whereas our sole requirement on $\gamma_{k+1}$ is the standard Assumption 5. However, it should be noted that the bounds in [44] are *non-asymptotic* (albeit doubly-exponential in $N$ and exponential in $k$), while we can only handle asymptotic convergence.

*Remark.* It is worth mentioning that Corollary 2 can be straightforwardly extended to the *Stochastic Particle-Optimization Sampling* algorithm of Zhang et al. [53], which is identical to SVGD but includes a constant diffusion term $\sigma > 0$. Since the analysis remains the same, we omit the details.

### 4.3 Two-Player Zero-sum Continuous Games

Min-max learning appears in several important machine learning tasks such as Generative Adversarial Networks [18] and adversarial training [35]. These learning problems can be formulated as a continuous zero-sum game between a min-player and a max-player. The min-player selects strategies from the set $\mathcal{X} \subset \mathbb{R}^d$, while the max-player selects strategies from the set $\mathcal{Y} \subset \mathbb{R}^d$, with the goal of finding a saddle point of a function $K(x, y)$:

$$\min_{x \in \mathcal{X}} \max_{y \in \mathcal{Y}} K(x, y). \tag{14}$$

However, solving (14) becomes challenging and sometimes impossible when $K$ is non-convex in $x$ and non-concave in $y$, as a solution to (14) may not even exist. To address this issue, *mixed Nash equilibriums* (MNEs) are introduced, where the pure strategies are replaced by probability distributions over the sets of strategies, which exist under mild assumptions on $K$ [13, 17].

Specifically, an MNE is represented by a pair of measures $(\mu^\star, \nu^\star) \in \mathscr{P}_2(\mathcal{X}) \times \mathscr{P}_2(\mathcal{Y})$, forming a saddle point of the functional $E(\mu, \nu) := \iint K(x, y)\mu(dx)\nu(dy)$. The quest for efficient solutions to the MNE problem in machine learning has led to the development of particle-based methods that offer approximate solutions [13, 20, 34]. However, the existing studies have primarily focused on analyzing the *continuous-time* dynamics of these methods. Below, we shift our focus to the more practical setting of the *discrete-time* system of interacting particles in these work:

$$\begin{cases} x_{k+1}^i = x_k^i - \gamma_{k+1} \frac{1}{N} \sum_j \nabla_x K(x_k^i, y_k^j) + \sqrt{2\tau\gamma_{k+1}}\, \xi_{k+1}^i \\ y_{k+1}^i = y_k^i + \alpha\gamma_{k+1} \frac{1}{N} \sum_j \nabla_y K(x_k^j, y_k^i) + \sqrt{2\alpha\tau\gamma_{k+1}}\, \zeta_{k+1}^i \end{cases}, \tag{GDA$_k$}$$

where $\xi_{k+1}^i$ and $\zeta_{k+1}^i$ are independent standard Gaussians, $\tau > 0$ is a hyperparameter chosen by the user, and $\alpha$ is the *scale* difference between the two players [34]. We will additionally consider the *optimistic* version of (GDA$_k$) in [20], which has shown empirical benefits over (GDA$_k$):

$$\begin{cases} x_{k+1}^i = x_k^i - \gamma_{k+1} \frac{1}{N} \sum_j \left( 2\nabla_x K(x_k^i, y_k^j) - \nabla_x K(x_{k-1}^i, y_{k-1}^j) \right) + \sqrt{2\tau\gamma_{k+1}}\, \xi_{k+1}^i \\ y_{k+1}^i = y_k^i + \alpha\gamma_{k+1} \frac{1}{N} \sum_j \left( 2\nabla_y K(x_k^j, y_k^i) - \nabla_y K(x_{k-1}^j, y_{k-1}^i) \right) + \sqrt{2\alpha\tau\gamma_{k+1}}\, \zeta_{k+1}^i. \end{cases} \tag{OGDA$_k$}$$

At first glance, it may seem that (GDA$_k$) and (OGDA$_k$) are distinct algorithms that cannot be analyzed together. However, our subsequent corollary reveals that these systems actually converge to the *same*

continuous-time dynamics:

$$\begin{cases} dX_t^i = -\frac{1}{N} \sum_j \nabla_x K(X_t^i, Y_t^j) \, dt + \sqrt{2\tau} dW_t^i \\ dY_t^i = \frac{\alpha}{N} \sum_j \nabla_y K(X_t^j, Y_t^i) \, dt + \sqrt{2\alpha\tau} dB_t^i \end{cases}, \qquad (\text{GDA}_t)$$

where $W_t^i$ and $B_t^j$ are collections of i.i.d. Brownian motions. The key enabling factor for this unified treatment is to allow for a *non-zero bias* in the algorithmic template (SAA).

To facilitate the analysis, we introduce a pairing of players' particles by setting $Q_t^i := (X_t^i, Y_t^i) \in \mathbb{R}^{2d}$, and define the drift $b : \mathbb{R}^{2d} \times \mathscr{P}_2(\mathbb{R}^{2d}) \to \mathbb{R}^{2d}$ as

$$b(q, \mu) := \int \begin{pmatrix} -\nabla_x K(q_1, q_2') \\ \alpha \nabla_y K(q_1', q_2) \end{pmatrix} \mu(dq'), \quad q = (q_1, q_2),$$

and the diffusion $\sigma(q, \mu) := \sqrt{2\tau} \, \mathrm{diag}(I_{d \times d}, \sqrt{\alpha} \, I_{d \times d})$. By doing so, we can then cast ($\text{GDA}_t$) in template of (Sys-SDE).

**Corollary 3.** *Assume that (1) $\nabla_x K$ and $\nabla_y K$ are Lipschitz, and (2) $\nabla_x K$ and $-\nabla_y K$ are dissipative, or (2') the domains $\mathcal{X}$ and $\mathcal{Y}$ are bounded. Then the algorithms ($\text{GDA}_k$) and ($\text{OGDA}_k$) converge in $\mathcal{W}_2$ to the same limits of ($\text{GDA}_t$).*

We note that the assumptions in (3) are mild and are even required for analying the *continuous-time* dynamics; see e.g., [34]. Consequently, our Corollary 3 provides a rigorous foundation for the consideration of continuous-time dynamics in existing studies such as [13, 20, 34].

*Remark.* The schemes ($\text{GDA}_k$) and ($\text{OGDA}_k$) above rely on *simultaneous* updates, i.e., from $(x_k, y_k)$, one obtains $(x_{k+1}, y_{k+1})$. However, empirical evidence suggests that *alternating* updates i.e., following $(x_k, y_k) \to (x_{k+1}, y_k) \to (x_{k+1}, y_{k+1})$, often preforms better. Our framework allows for this flexibility, as it is easy to cast the alternating ($\text{GDA}_k$) and ($\text{OGDA}_k$) as stochastic approximation schemes satisfying Assumption 4, see [24, Proposition 4] for an example of this argument.

## 4.4 Kinetic Equations

In this section, we study the *kinetic equations* defined on the space of probability measures. Initially emerging from the physics community, these equations have recently gained attention in the machine learning community due to their connection to *Wasserstein gradient flows* [15, 32, 39, 40, 48, 49, 52]. Here, we denote $\rho$ as a probability density, and we examine three distinct "energy" functionals: an internal energy $\mathcal{U}$, a potential energy $\mathcal{V}$, and an interaction energy $\mathcal{W}$. These functionals are defined as follows:

$$\mathcal{U}(\rho) = \int U(\rho(x)) \, dx, \quad \mathcal{V}(\rho) = \int V(x) \, \rho(dx), \quad \mathcal{W}(\rho) = \frac{1}{2} \iint W(x - y) \, \rho(dx) \rho(dy). \quad (15)$$

The most interesting scenario is when $U(s) = s \log s$ represents the *entropy*, in which case the Wasserstein gradient flow with finite particles becomes [5, 7]:

$$dX_t^i = -\nabla V(X_t^i) \, dt - \frac{1}{N} \sum_{j=1}^N \nabla W(X_t^i - X_t^j) \, dt + \sqrt{2} \, dW_t^i, \quad i = 1, \dots, N. \quad (\text{Kin-SDE})$$

By setting $b(x, \mu) := -\nabla V(x) - (\nabla W * \mu)(x)$ and $\sigma(x, \mu) \equiv \sqrt{2}$, we again see that (Kin-SDE) is a special case of (Sys-SDE).

In the physics community, the equation is commonly simulated using a system of interacting particles through the *proximal point method* [6, 48]:

$$x_{k+1}^i = x_k^i - \gamma_{k+1} \left( \nabla V(x_{k+1}^i) + \frac{1}{N} \sum_{j=1}^N \nabla W(x_{k+1}^i - x_k^j) \right) + \sqrt{2\gamma_{k+1}} \, \xi_{k+1}^i. \quad (\text{Kin-Prox})$$

Note that the right-hand side of (Kin-Prox) involves the next iterates $x_{k+1}^i$ so that, as opposed to the simple Euler discretization, it is an *implicit* rule. The key factor that enables the application of our framework to these implicit schemes is the observation that (Kin-Prox) can be formulated as a stochastic approximation scheme in (SAA) by incorporating a *non-zero bias* term; see Appendix C. Our next result provides a rigorous guarantee for these methods:

**Corollary 4.** *Assume that (1) $V$ is dissipative and $\nabla V$ is $L$-Lipschitz, (2) $W$ is symmetric and $\nabla W$ is $L$-Lipschitz, (3) There exists some $M_W \geq 0$ such that for all $x, y \in \mathbb{R}^d$, $\langle \nabla W(x) - \nabla W(y), x - y \rangle \geq -M_W$. Then, the iterates (Kin-Prox) converge in $\mathcal{W}_2$ to the same limit sets as (Kin-SDE).*

We remark that the convergence of (Kin-Prox) under these standard assumptions is known [15, 48]. However, we emphasize that the aforementioned results only apply to *deterministic* updates, while our proof is robust enough to accommodate updates with *finite-variance noise*. This flexibility sets our approach apart, allowing for a broader range of practical applications.

## 5    Conclusion and Future Directions

In conclusion, our work has successfully bridged the gap between continuous- and discrete-time schemes by establishing the convergence of discrete-time algorithms to their continuous-time counterparts, drawing inspiration from dynamical system theory. This achievement offers a flexible framework that can accommodate practical schemes and unify tasks from various domains, including machine learning, game theory, and physics. By introducing a comprehensive framework for analyzing stochastic approximation schemes, providing convergence proofs, and presenting easily verifiable conditions at the finite particle level, our contributions enhance the understanding and application of stochastic approximation schemes for simulating interacting particle systems.

In our future works, we will explore the exciting possibilities offered by the dynamical system theory of [4] to derive convergence rates through $\lambda$-pseudotrajectories. This avenue of research will allow us to establish exponential convergence for dynamics using smaller step-sizes after a "burn-in" time, thereby sharpening our understanding for the long-term behavior of these practical schemes.

Additionally, we aim to relax the Lipschitzness assumption on the drift and expand the scope of guaranteed algorithms, such as the Ensemble Kalman Sampler [16]. We also aim to explore further applications in natural sciences, including stochastic mean-field FitzHugh-Nagumo models and networks of Hodgkin-Huxley neurons [1]. These models are important because they provide a mathematical description of neuronal dynamics, contribute to our understanding of neurological disorders, and inform the development of brain-computer interfaces. By leveraging our novel framework, we hope to offer rigorous guarantees for the algorithms employed in these domains while also designing new, efficient approaches that can support computational neuroscience research.

## Acknowledgments and Disclosure of Funding

This work was supported by the European Research Council (ERC) under the European Union's Horizon 2020 research and innovation program grant agreement No 815943. YPH acknowledges funding through an ETH Foundations of Data Science (ETH-FDS) postdoctoral fellowship.

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

# A Properties of Mean-field Transfer to Particle System

**Lemma A.1.** *The Assumptions 1–3 transfer seamlessly to the aggregated notions of drift and diffusion given in* (7):

*1. If $b(\cdot, \cdot)$ and $\sigma(\cdot, \cdot)$ satisfy Assumption 1, then $\boldsymbol{b}(\cdot)$ and $\boldsymbol{\sigma}(\cdot)$ are $L(\sqrt{N} + 1)$-Lipschitz.*

*2. If $b(\cdot, \cdot)$ satisfies Assumption 2, then $\boldsymbol{b}(\cdot)$ satisfies the same condition with the constant $\sqrt{N}C_v$.*

*3. If $\sigma(\cdot, \cdot)$ satisfies Assumption 3, then the same holds for $\boldsymbol{\sigma}(\cdot)$ with constant $\sqrt{N}K$.*

*Proof.* We prove these statements separately:

**1.** Let $x, y \in (\mathbb{R}^d)^{\otimes N}$ and define $a_i = |x^i - y^i|$. First, notice that $\mathcal{W}_2^2(\mu_x, \mu_y) \le \frac{1}{N} \sum a_i^2$, as the average on the right-hand-side corresponds to the specific coupling of $x_i \leftrightarrow y_i$. Now, observe that

$$
\begin{aligned}
|\boldsymbol{b}(x) - \boldsymbol{b}(y)|^2 &= \sum_i |b(x^i, \mu_x) - b(y^i, \mu_y)|^2 \\
&\le L^2 \sum_i \left(a_i + \mathcal{W}_2(\mu_x, \mu_y)\right)^2 \\
&\le L^2 \sum_i \left(a_i + \sqrt{\frac{1}{N} \sum_j a_j^2}\right)^2.
\end{aligned}
$$

Let $\boldsymbol{a} = (a_1, \ldots, a_N)$, and notice that the last quantity above is equal to

$$
L^2 \left| \boldsymbol{a} + \sqrt{\tfrac{1}{N}} |\boldsymbol{a}| \cdot \mathbf{1} \right|^2 = L^2 |\boldsymbol{a}|^2 \left| \frac{\boldsymbol{a}}{|\boldsymbol{a}|} + \sqrt{\tfrac{1}{N}} \cdot \mathbf{1} \right|^2 \le L^2 |\boldsymbol{a}|^2 N \left(1 + \sqrt{\tfrac{1}{N}}\right)^2.
$$

This means that

$$
|\boldsymbol{b}(x) - \boldsymbol{b}(y)| \le L(\sqrt{N} + 1)|x - y|.
$$

For the diffusion, it suffices to notice that

$$
\|\boldsymbol{\sigma}(x) - \boldsymbol{\sigma}(y)\|_{\mathrm{F}}^2 = \sum_i \|\sigma(x^i, \mu_x) - \sigma(y^i, \mu_y)\|_{\mathrm{F}}^2.
$$

The rest of the proof is similar to the one for the drift.

**2.** We have

$$
\frac{1}{N} \langle x, \boldsymbol{b}(x) \rangle = \frac{1}{N} \sum_{i=1}^N \langle x^i, b(x^i, \mu_x) \rangle \le C_v \left(\frac{1}{N} \sum |x^i| + 1\right) \le C_v \left(\frac{1}{\sqrt{N}} \sqrt{\sum |x^i|^2} + 1\right),
$$

where in the last inequality, we used Cauchy-Schwarz. This implies $\langle x, \boldsymbol{b}(x) \rangle \le C_v \sqrt{N}(|x| + 1)$.

**3.** It is easy to see that

$$
\|\boldsymbol{\sigma}(x)\|_F^2 = \mathrm{tr}(\boldsymbol{\sigma}(x)^\top \boldsymbol{\sigma}(x)) = \sum_{i=1}^N \mathrm{tr}(\sigma(x^i, \mu_x)^\top \sigma(x^i, \mu_x)) \le NK^2. \qquad \blacksquare
$$

**Lemma A.2.** *If $b(\cdot, \cdot)$ is $(\alpha, \beta)$-dissipative on average, then $\boldsymbol{b}(\cdot)$ is $(\alpha, N\beta)$-dissipative in the usual sense, that is, for all $x \in (\mathbb{R}^d)^{\otimes N}$, $\langle x, \boldsymbol{b}(x) \rangle \le -\alpha|x|^2 + N\beta$.*

*Proof.* Observe that for $x \in (\mathbb{R}^d)^{\otimes N}$ we have

$$
\begin{aligned}
\frac{1}{N} \langle x, \boldsymbol{b}(x) \rangle &= \frac{1}{N} \sum_{i=1}^N \langle x^i, b(x^i, \mu_x) \rangle = \mathbb{E}_{\mu_x}[\langle y, b(y, \mu_x) \rangle] \le -\alpha \mathbb{E}_{\mu_x} |y|^2 + \beta \\
&= -\alpha \frac{1}{N} \sum_{i=1}^N |x^i|^2 + \beta = -\alpha \frac{1}{N} |x|^2 + \beta.
\end{aligned}
$$

This means that $\langle x, \boldsymbol{b}(x) \rangle \le -\alpha|x|^2 + N\beta$. $\qquad \blacksquare$

# B  The Main Theorem

## B.1  Proof of Theorem 1

Recall the *Picard process* (Picard):

$$\Pi_s^{(t)} = X_t + \int_0^s \boldsymbol{b}(X_{t+u})\, du + \int_0^s \boldsymbol{\sigma}(X_{t+u})\, dW_u^{(t)}.$$

We break down the proof into four steps: first, we prove that the Picard process is close to the flow (Flow), and then we bound the distance between the Picard process and the interpolation (Int). We then conclude the proof of the WAPT property. Lastly, we prove that stability is implied by dissipativity.

### § Distance of Picard from Flow.

$$
\begin{aligned}
\mathbb{E}|\Pi_s^{(t)} - \Phi_s^{(t)}|^2 &\le 2\,\mathbb{E}\left|\int_0^s \boldsymbol{b}(X_{t+u}) - \boldsymbol{b}(\Phi_u^{(t)})\, du\right|^2 + 2\,\mathbb{E}\left|\int_0^s \boldsymbol{\sigma}(X_{t+u}) - \boldsymbol{\sigma}(\Phi_u^{(t)})\, dW_u^{(t)}\right|^2 \\
&\le T \int_0^s \mathbb{E}|\boldsymbol{b}(\Phi_u^{(t)}) - \boldsymbol{b}(X_{t+u})|^2\, du + 2\,\mathbb{E}\int_0^s \|\boldsymbol{\sigma}(X_{t+u}) - \boldsymbol{\sigma}(\Phi_u^{(t)})\|_{\mathrm{F}}^2\, du \\
&\le 2(T+1)L^2 \int_0^s \mathbb{E}|\Phi_u^{(t)} - X_{t+u}|^2
\end{aligned}
$$

where we used Lipschitzness of $\boldsymbol{b}$ and $\boldsymbol{\sigma}$ (implied by Assumption 1 and Lemma A.1), Itô's isometry (see, e.g., [54, Lemma 3.4]), and Lemma A.1.

### § Distance of Picard to Interpolation.
We place a bar above a symbol to denotes its piecewise constant interpolation.

$$
\begin{aligned}
\mathbb{E}|\Pi_s^{(t)} - X_{t+s}|^2 &= \mathbb{E}\left|\int_t^{t+s} \boldsymbol{b}(X_u) - \boldsymbol{b}(\overline{X_u})\, du + \int_t^{t+s} \boldsymbol{\sigma}(X_u) - \boldsymbol{\sigma}(\overline{X_u})\, dW_u^{(t)} + \Delta_P(t,s)\right|^2 \\
&\le 3(T+1)L^2 \int_t^{t+s} \mathbb{E}|X_u - \overline{X_u}|^2\, du + 3\,\mathbb{E}\left|\Delta_P(t,s)\right|^2,
\end{aligned}
$$

where $\Delta_P(t,s)$ is the accumulated noise and bias from time $t$ to time $t+s$, which is equal to

$$\Delta_P(t,s) := \sum_{i=k}^{n-1} \gamma_{i+1} P_{i+1} + (t+s-\tau_n)\,\mathbb{E}[P_{n+1} \mid \mathscr{F}_{t+s}] - (t-\tau_k)\,\mathbb{E}[P_{k+1} \mid \mathscr{F}_t], \tag{B.1}$$

with $n = m(t+s)$ and $k = m(t)$. It is shown in [23] that $\lim_{t\to\infty} \mathbb{E}\left|\Delta_P(t,s)\right|^2 = 0$, a.s.

Continuing to bound the inside of the integral, we have

$$\mathbb{E}|X_t - x_k|^2 \le 3(t-\tau_k)^2 (\mathbb{E}|\boldsymbol{b}(x_k)|^2 + \mathbb{E}|P_{k+1}|^2) + 3(t-\tau_k)\,\mathbb{E}\,\mathrm{tr}(\boldsymbol{\sigma}(x_k)^\top \boldsymbol{\sigma}(x_k))$$

where we used the fact that conditional expectation is a contraction in $L^2$, and

$$
\begin{aligned}
\mathbb{E}|\boldsymbol{\sigma}(x_k)\,\xi_{k+1}|^2 &= \mathbb{E}\,\mathrm{tr}\big(\xi_{k+1}^\top \boldsymbol{\sigma}(x_k)^\top \boldsymbol{\sigma}(x_k)\,\xi_{k+1}\big) \\
&= \mathbb{E}\,\mathrm{tr}\big(\boldsymbol{\sigma}(x_k)^\top \boldsymbol{\sigma}(x_k)\,\xi_{k+1}\xi_{k+1}^\top\big) \\
&= \mathbb{E}\,\mathrm{tr}\big(\boldsymbol{\sigma}(x_k)^\top \boldsymbol{\sigma}(x_k)\,\mathbb{E}[\xi_{k+1}\xi_{k+1}^\top \mid \mathscr{F}_{\tau_k}]\big) \\
&= \mathbb{E}\,\mathrm{tr}\big(\boldsymbol{\sigma}(x_k)^\top \boldsymbol{\sigma}(x_k)\big).
\end{aligned}
$$

Moreover, by Assumption 3 we have $\mathbb{E}\,\mathrm{tr}(\boldsymbol{\sigma}(x_k)^\top \boldsymbol{\sigma}(x_k)) = \mathcal{O}(1)$. We thus get by Lemma B.1

$$\mathbb{E}|X_t - x_k|^2 \le 3C\gamma_{k+1}^2 (1/\gamma_{k+1} + 1) + 3C\gamma_{k+1} = \mathcal{O}(\gamma_{k+1}).$$

This implies

$$\sup_{s\in[0,T]} \mathbb{E}|\Pi_s^{(t)} - X_{t+s}|^2 \le CT^2 L^2 \sup_{t\le u\le t+T} \overline{\gamma_u} + 3\,\mathbb{E}\left|\Delta_P(t,T)\right|^2 =: A_t,$$

with $A_t \to 0$ as $t \to \infty$, a.s.

§ **Concluding the proof of APT.** By Grönwall inequality,

$$\mathbb{E}|X_{t+s} - \Phi_s^{(t)}|^2 \le C \int_0^s \mathbb{E}|X_{t+u} - \Phi_u^{(t)}|^2 + A_t \le A_t \exp(sC) \le A_t \exp(TC) \to 0$$

as $t \to \infty$. Since

$$\mathcal{W}_2^2(\text{law}(X_{t+s}), \text{law}(\Phi_s^{(t)})) \le \mathbb{E}|X_{t+s} - \Phi_s^{(t)}|^2,$$

we get the desired result.

§ **Stability.** Lemma A.2 implies that under dissipativity on average, the iterates are stable, and as in [23, Theorem 3], we get the desired convergence result.

## B.2 On Propagation of Chaos

Theorem 1 shows that the law $\mu_k$ of $x_k \in (\mathbb{R}^d)^{\otimes N}$ converges in the Wasserstein space to the limit-set (or the *internally chain-transitive* (ICT) set) $S \subset \mathscr{P}_2((\mathbb{R}^d)^{\otimes N})$ of the corresponding flow:

$$\lim_{k \to \infty} \inf_{\mu \in S} \mathcal{W}_2(\mu_k, \mu) = 0.$$

By looking only at the first particle of $x_k$, namely, $x_k^1$, and given that the dynamics is exchangeable, it follows that

$$\begin{aligned}
\mathcal{W}_2^2(\mu_k, \mu) &= \inf_\pi \int |x - y|^2 \, \pi(dx, dy) \\
&= \inf_\pi \left( \int |x^1 - y^1|^2 \, \pi^1(dx^1, dy^1) + \cdots + \int |x^N - y^N|^2 \, \pi^N(dx^N, dy^N) \right) \\
&\ge \inf_{\pi^1} \int |x^1 - y^1|^2 \, \pi^1(dx^1, dy^1) \\
&= \mathcal{W}_2^2(\text{law}(x_k^1), \text{marginal}_1(\mu)),
\end{aligned}$$

where $\pi^1(dx^1, dy^1) = \int \pi(x, y) \, dx^2 dy^2 \cdots dx^N dy^N$, and we used the exchangability in deducing that the law of $y^i$ are the same as the first marginal of $\mu$, for all $i = 1, \ldots, N$. Hence, as the limit-set $S'$ of the first component of the SDE $X_t^1$ is a subset of the marginal of $S$,

$$\lim_{k \to \infty} \inf_{\nu \in S'} \mathcal{W}_2(\text{law}(x_k^1), \nu) \le \lim_{k \to \infty} \inf_{\nu \in \text{marginal}_1(S)} \mathcal{W}_2(\text{law}(x_k^1), \nu) = 0.$$

This means that the first particle converges in law to the ICT sets of the corresponding SDE. Assuming a uniform propagation of chaos, we also know that the law of $X_t^1$ has a distance of $O(1/N)$ from the mean-field equation, and hence, we get that the law of the particles following the discrete algorithm have controllable distance from the mean-field dynamics.

## B.3 Supporting Lemmas

**Lemma B.1.** *Suppose Assumptions 1–5 hold. One has* $\mathbb{E}|\boldsymbol{b}(x_k)|^2 = \mathcal{O}(1/\gamma_{k+1})$, $\mathbb{E}|\varepsilon_{k+1}|^2 = \mathcal{O}(\gamma_{k+1})$, *and* $\mathbb{E}|P_{k+1}|^2 = \mathcal{O}(1)$.

*Proof.* We repeatedly use the fact that $\mathbb{E}|\boldsymbol{b}(x_k)|^2 \le 2L^2 \mathbb{E}|x_k|^2 + \mathbb{E}|\boldsymbol{b}(x_0)|^2 =: 2L^2 \mathbb{E}|x_k|^2 + C_0$. By Assumption 4, $\mathbb{E}|\varepsilon_{k+1}|^2 \le \mathcal{O}(\gamma_{k+1}^2)a_k + \mathcal{O}(\gamma_{k+1})$, and we have

$$\mathbb{E}|P_{k+1}|^2 \le 2\mathbb{E}|\varepsilon_{k+1}|^2 + 2\mathbb{E}|U_{k+1}|^2 = \mathcal{O}(\gamma_{k+1}^2)a_k + \mathcal{O}(1). \tag{B.2}$$

Moreover, as $\sqrt{p+q} \le \sqrt{p} + \sqrt{q}$, we have

$$\sqrt{\mathbb{E}|P_{k+1}|^2} \le \mathcal{O}(\gamma_{k+1})\sqrt{a_k} + \mathcal{O}(1). \tag{B.3}$$

Assumption 3 also implies that $\mathbb{E}|\boldsymbol{\sigma}(x_k)\xi_{k+1}|^2 \le C_\sigma$.

Define $a_k := \mathbb{E}|x_k|^2$. Then,

$$a_{k+1} - a_k = \gamma_{k+1}^2 \, \mathbb{E}|\boldsymbol{b}(x_k) + P_{k+1}|^2 + \gamma_{k+1} \, \mathbb{E}|\boldsymbol{\sigma}(x_k)\xi_{k+1}|^2 + 2\gamma_{k+1} \, \mathbb{E}\langle x_k, \boldsymbol{b}(x_k) + P_{k+1}\rangle$$

$$+ 2\gamma_{k+1}^{1/2} \, \mathbb{E}\langle x_k, \boldsymbol{\sigma}(x_k)\xi_{k+1}\rangle + 2\gamma_{k+1}^{3/2} \, \mathbb{E}\langle \boldsymbol{b}(x_k) + P_{k+1}, \boldsymbol{\sigma}(x_k)\xi_{k+1}\rangle$$

$$\leq 2L^2\gamma_{k+1}^2 a_k + \gamma_{k+1}^2 C_0 + 2\gamma_{k+1}^2 \, \mathbb{E}|P_{k+1}|^2 + \gamma_{k+1} C_\sigma + 2\gamma_{k+1} C_v(\sqrt{a_k} + 1)$$

$$+ 2\gamma_{k+1}\sqrt{a_k}\sqrt{\mathbb{E}|P_{k+1}|^2} + 2\gamma_{k+1}^{3/2}\sqrt{C_\sigma}\sqrt{\mathbb{E}|P_{k+1}|^2} \tag{B.4}$$

Plugging the bounds from (B.2) and (B.3) into (B.4) gives

$$a_{k+1} - a_k \leq \mathcal{O}(\gamma_{k+1}^2)a_k + \mathcal{O}(\gamma_{k+1})\sqrt{a_k} + \mathcal{O}(\gamma_{k+1})$$

$$=: P\gamma_{k+1}^2 a_k + Q\gamma_{k+1}\sqrt{a_k} + R\gamma_{k+1},$$

for some $P, Q, R > 0$ that do not depend on $k$.

We now prove $a_k \leq M/\gamma_{k+1}$ for some fixed $M > 0$ via induction. Suppose this is the case for $k$. For $k + 1$ we have

$$a_{k+1} \leq (P\gamma_{k+1}^2 + 1)a_k + Q\gamma_{k+1}\sqrt{a_k} + R\gamma_{k+1}$$

$$\leq M(P\gamma_{k+1} + 1/\gamma_{k+1}) + \sqrt{M}Q\sqrt{\gamma_{k+1}} + R\gamma_{k+1}$$

$$\overset{!}{\leq} M/\gamma_{k+2}.$$

The last inequality is equivalent to the fact that the following quadratic equation (in $\sqrt{M}$) has a bounded largest root (and the bound shall not depend on $k$):

$$M(P\gamma_{k+1} + 1/\gamma_{k+1} - 1/\gamma_{k+2}) + \sqrt{M}Q\sqrt{\gamma_{k+1}} + R\gamma_{k+1}$$

Notice that by Assumption 5, the leading coefficient is negative, and the larger root is computed as

$$\frac{Q\sqrt{\gamma_{k+1}} + Q\sqrt{\gamma_{k+1}} + \sqrt{4R(\gamma_{k+1}/\gamma_{k+2} - P\gamma_{k+1}^2 - 1)}}{2(1/\gamma_{k+2} - P\gamma_{k+1} - 1/\gamma_{k+1})}$$

$$\leq \frac{2Q\sqrt{\gamma_{k+1}} + 2\sqrt{R}\sqrt{\gamma_{k+1}/\gamma_{k+2}}}{2\gamma_{k+1}/\gamma_{k+2}} \leq 2Q\sqrt{\gamma_{k+2}} + \sqrt{R}\sqrt{\gamma_{k+2}/\gamma_{k+1}} < 2Q + \sqrt{R} =: M.$$

The claim for $\boldsymbol{b}$ follows by Lipschitzness, from which the claim for the bias and perturbation follows directly. ∎

## C  Proofs of Results for Applications

### C.1  Two-Layer Neural Networks and Mean-field Langevin

*Proof of Corollary 1.* **Smoothness of drift:** We start by showing Lipschitzness with respect to the measure parameter of the drift. First, observe that $\nabla_\theta W(\theta, \cdot)$ is Lipschitz:

$$|\nabla_\theta W(\theta, \theta') - \nabla_\theta W(\theta, \theta'')| = |\mathbb{E}_{z\sim\mathcal{D}}[(\varphi(z, \theta') - \varphi(z, \theta''))\nabla_\theta\varphi(z, \theta)]|$$

$$= |\mathbb{E}_{z\sim\mathcal{D}}[(\kappa(\langle z, \theta'\rangle) - \kappa(\langle z, \theta''\rangle))\kappa'(\langle z, \theta\rangle)z]|$$

$$\leq C|\theta' - \theta''|,$$

due to the boundedness of $\kappa'$ and $z$, and Lipschitzness of $\kappa$.

Now, consider $\mu, \nu \in \mathscr{P}_2(\mathbb{R}^d)$ and let $\pi$ be the optimal coupling (in $\mathcal{W}_2$ sense) between them. Then, for a fixed $\theta \in \mathbb{R}^d$,

$$|b(\theta, \mu) - b(\theta, \nu)|^2 = \left|\int \nabla_\theta W(\theta, p) - \nabla_\theta W(\theta, q)\,\pi(dp, dq)\right|^2$$

$$\leq \int |\nabla_\theta W(\theta, p) - \nabla_\theta W(\theta, q)|^2\,\pi(dp, dq)$$

$$\leq \int C^2|p - q|^2\,\pi(dp, dq)$$

$$= C^2 W_2^2(\mu, \nu).$$

Next, we show for a fixed measure $\mu \in \mathscr{P}_2(\mathbb{R}^d)$, $b(\cdot, \mu)$ is Lipschitz in the first input.

$$|b(\theta, \mu) - b(\theta', \mu)| \le \left| \int \nabla_\theta W(\theta, p) - \nabla_\theta W(\theta', p) \, \mu(dp) \right| + |\nabla V(\theta) - \nabla V(\theta')|.$$

Let us treat each term separately. We have

$$
\begin{aligned}
|\nabla_\theta W(\theta, p) - \nabla_\theta W(\theta', p)| &= |\mathbb{E}_{z \sim \mathcal{D}}[\varphi(z, p)(\nabla_\theta \varphi(z, \theta) - \nabla_\theta \varphi(z, \theta'))]| \\
&= |\mathbb{E}_{z \sim \mathcal{D}}[\kappa(\langle p, z \rangle)(\kappa'(\langle z, \theta \rangle) - \kappa'(\langle z, \theta' \rangle)) z]| \\
&\le C \, \mathbb{E}_{z \sim \mathcal{D}}[|z||\theta - \theta'| z] \\
&\le C|\theta - \theta'|.
\end{aligned}
$$

Similarly,

$$|\nabla V(\theta) - \nabla V(\theta')| = \left| \mathbb{E}_{(y,z) \sim \mathcal{D}}[yz(\kappa'(\langle \theta, z \rangle) - \kappa'(\langle \theta', z \rangle)) \mathcal{D}(dy, dz)] \right| \le C|\theta - \theta'|.$$

Thus,

$$|b(\theta, \mu) - b(\theta', \nu)| \le |b(\theta, \mu) - b(\theta, \nu)| + |b(\theta, \nu) - b(\theta', \nu)| \le L(|\theta - \theta'| + \mathcal{W}_2(\mu, \nu)),$$

showing $b$ satisfies Assumption 1.

**Growth control:** First, let us calculate

$$
\begin{aligned}
\int \langle \theta, b(\theta, \mu) \rangle \, \mu(d\theta) &= \iint \varphi(z, \theta') \langle \theta, \nabla_\theta \varphi(z, \theta) \rangle \, \mathcal{D}(dz)\mu(d\theta')\mu(d\theta) \\
&\quad - \iint y \langle \theta, \nabla_\theta \varphi(z, \theta) \rangle \, \mathcal{D}(dy, dz)\mu(d\theta) \\
&\quad - \lambda \int |\theta|^2 \, \mu(d\theta) \\
&= \iint \varphi(z, \theta') \kappa'(\langle z, \theta \rangle) \langle \theta, z \rangle \, \mathcal{D}(dz)\mu(d\theta')\mu(d\theta) \\
&\quad - \iint y \, \kappa'(\langle z, \theta \rangle) \langle \theta, z \rangle \, \mathcal{D}(dy, dz)\mu(d\theta) \\
&\quad - \lambda \int |\theta|^2 \, \mu(d\theta).
\end{aligned}
$$

As $\varphi$, $\kappa'$, and $\mathrm{supp}(\mathcal{D})$ are bounded, we can see that

$$\left| \int \langle \theta, b(\theta, \mu) \rangle \, \mu(d\theta) \right| \le C \iint |\theta||z| \, \mathcal{D}(dz)\mu(d\theta) + C' \int |\theta||z| \, \mathcal{D}(dz)\mu(d\theta) \le C \int |\theta| \, \mu(d\theta),$$

thus, satisfying Assumption 2.

**Dissipativity on average:** Here we use the extra assumption that $|a \, \kappa'(a)|$ is bounded. We directly bound the terms $\kappa'(\langle \theta, z \rangle) \langle \theta, z \rangle$ above and obtain

$$\int \langle \theta, b(\theta, \mu) \rangle \, \mu(d\theta) \le -\lambda \int |\theta|^2 \, \mu(d\theta) + C. \qquad \blacksquare$$

### C.2 Stein Variational Gradient Descent

*Proof of Corollary 2.* While the first term in the drift is standard to work with (see Section 4.4), it is the second term in the drift that makes it difficult to analyze. Specifically, we prove the dissipativity on average only for empirical measures. While this would be enough for our purposes (and Theorem 1 goes through), it is an interesting future direction to see when does dissipativity hold in a more general setup. Moreover, for simplicity, we only consider the case where the kernel $K$ is of the form $K(x, y) = h(x - y)$, for some function $h$.

Below, we first prove that $b$ is dissipative on average, which implies that the law of the iterates will be in a compact subset of $\mathscr{P}_2(\mathbb{R}^d)$. Then, we show that $b$ is smooth on this compact subset.

**Dissipativity on average:** Due to $K$ being symmetric, $\nabla_2 K(x, y) = -\nabla_2 K(y, x)$. We thus have

$$\iint \langle x - y, \nabla_2 K(x, y) \rangle \, \mu(dx)\mu(dy)$$

$$= \iint \langle x, \nabla_2 K(x, y) \rangle \, \mu(dx)\mu(dy) - \iint \langle y, \nabla_2 K(x, y) \rangle \, \mu(dx)\mu(dy)$$

$$= \iint \langle x, \nabla_2 K(x, y) \rangle \, \mu(dx)\mu(dy) + \iint \langle y, \nabla_2 K(y, x) \rangle \, \mu(dx)\mu(dy)$$

$$= 2 \iint \langle x, \nabla_2 K(x, y) \rangle \, \mu(dx)\mu(dy).$$

Thus,

$$\iint \langle x, \nabla_2 K(x, y) \rangle \, \mu(dx)\mu(dy) = \frac{1}{2} \iint \langle x - y, \nabla_2 K(x, y) \rangle \, \mu(dx)\mu(dy) \le \eta$$

by Cauchy-Schwarz and the assumption that $|\nabla_2 K(x, y)| \le \eta/|x - y|$. With similar arguments, and using dissipativity of $V$, we have

$$\iint \langle x, \nabla V(y) \rangle K(x, y) \, \mu(dx)\mu(dy)$$

$$= \iint \langle x, \nabla V(x) \rangle K(x, y) \, \mu(dx)\mu(dy) - \frac{1}{2} \iint \langle x - y, \nabla V(x) - \nabla V(y) \rangle K(x, y) \, \mu(dx)\mu(dy)$$

$$\ge \alpha \iint |x|^2 K(x, y) \, \mu(dx)\mu(dy) - \beta \|K\|_\infty$$

$$- \frac{1}{2} L \iint |x - y|^2 K(x, y) \, \mu(dx)\mu(dy)$$

$$\ge \alpha \iint |x|^2 K(x, y) \, \mu(dx)\mu(dy) - \beta \|K\|_\infty + \frac{L\eta}{2}.$$

As $K(x, x) = h(0)$ and $K(x, y) > 0$ for all $x, y$, and that $\mu$ is an empirical measure $\mu = \frac{1}{N} \sum_{i=1}^N \delta_{x_i}$, the last quantity is equal to

$$\frac{1}{N^2} \sum_i |x_i|^2 \sum_j K(x_i, x_j) \ge \frac{1}{N^2} \sum_i |x_i|^2 K(x_i, x_i) \ge \frac{h(0)}{N} \int |x|^2 \, \mu(dx).$$

In total, we derive that $b$ is dissipative on average.

**Smoothness of the drift:** We have, for $\mu$ in a compact set of $\mathscr{P}_2(\mathbb{R}^d)$

$$|b(x, \mu) - b(x', \mu)|$$

$$\le \left| \int \nabla_2 K(x, y) - \nabla_2 K(x', y) \, \mu(dy) \right| + \left| \int (K(x, y) - K(x', y)) \nabla V(y) \, \mu(dy) \right|$$

$$\le L|x - x'| \left( 1 + \int |\nabla V(y)| \, \mu(dy) \right)$$

$$\le L|x - x'| \left( 1 + C \int (1 + |y|^2) \, \mu(dy) \right) < L'|x - x'|.$$

Moreover, take $\mu, \nu$ in the same compact set, and let $\pi$ be the optimal coupling (in $\mathcal{W}_2$ sense). Then,

$$|b(x, \mu) - b(x, \nu)|^2$$

$$\le 2 \left| \int \nabla_2 K(x, y) - \nabla_2 K(x, z) \, \pi(dy, dz) \right|^2$$

$$+ 2 \left| \int K(x, y) \nabla V(y) - K(x, z) \nabla V(z) \, \pi(dy, dz) \right|^2$$

$$\le 2 L^2 \mathcal{W}_2^2(\mu, \nu)$$

$$+ 2 \left| \int K(x, y) \nabla V(y) - K(x, z) \nabla V(y) + K(x, z) \nabla V(y) - K(x, z) \nabla V(z) \, \pi(dy, dz) \right|^2$$

$$\le 2 L^2 \mathcal{W}_2^2(\mu, \nu) + 4(L^2 + L'^2) \mathcal{W}_2^2(\mu, \nu). \qquad \blacksquare$$

### C.3 Two-player Zero-sum Continuous Games

*Proof of Corollary 3.* Recall that

$$b(q,\mu) := \int \begin{pmatrix} -\nabla_x K(q_1, q_2') \\ \alpha\,\nabla_y K(q_1', q_2) \end{pmatrix} \mu(dq'), \quad q = (q_1, q_2).$$

**Smoothness of drift:** For a fixed $\mu \in \mathscr{P}_2(\mathbb{R}^d)\mathbb{R}^{2d}$ and $q, r \in \mathbb{R}^{2d}$ we have

$$
\begin{aligned}
|b(q,\mu) - b(r,\mu)|^2 &\leq \int \left| \begin{pmatrix} -\nabla_x K(q_1, q_2') \\ \alpha\,\nabla_y K(q_1', q_2) \end{pmatrix} - \begin{pmatrix} -\nabla_x K(r_1, q_2') \\ \alpha\,\nabla_y K(q_1', r_2) \end{pmatrix} \right|^2 \mu(dq') \\
&\leq L^2 |q_1 - r_1|^2 + \alpha^2 L^2 |q_2 - r_2|^2 \\
&\leq L^2 |q - r|^2,
\end{aligned}
$$

where we used $L$-Lipschitzness of $\nabla_x K$ and $\nabla_y K$.

Now, for a fixed $q \in \mathbb{R}^{2d}$, and $\mu, \nu \in \mathscr{P}_2(\mathbb{R}^d)\mathbb{R}^{2d}$ with optimal coupling $\pi$, we have

$$
\begin{aligned}
|b(q,\mu) - b(q,\nu)|^2 &\leq \iint \left| \begin{pmatrix} -\nabla_x K(q_1, r_2) \\ \alpha\,\nabla_y K(r_1, q_2) \end{pmatrix} - \begin{pmatrix} -\nabla_x K(q_1, r_2') \\ \alpha\,\nabla_y K(r_1', q_2) \end{pmatrix} \right|^2 \pi(dr, dr') \\
&\leq L^2 \iint |r_2 - r_2'|^2 + |r_1 - r_1'|^2 \, \pi(dr, dr') \\
&= L^2 \mathcal{W}_2^2(\mu, \nu).
\end{aligned}
$$

**Average dissipativity of drift:** Suppose $\nabla_x K$ and $-\nabla_y K$ are $(a, \beta)$-dissipative. Then

$$
\begin{aligned}
\int \langle q, b(q,\mu) \rangle \, \mu(dq) &= \iint \langle q_1, -\nabla_x K(q_1, q_2') \rangle + \alpha \langle q_2, \nabla_y K(q_1', q_2) \rangle \, \mu(dq')\mu(dq) \\
&\leq \int -a\alpha(|q_1|^2 + |q_2|^2) \, \mu(dq) + 2\beta,
\end{aligned}
$$

implying that $b(\cdot, \cdot)$ is $(a\alpha, 2\beta)$-dissipative on average.

If, on the other hand, the domains $\mathcal{X}$ and $\mathcal{Y}$ are bounded, observe that by Cauchy-Schwarz

$$\left| \int \langle q, b(q,\mu) \rangle \, \mu(dq) \right| \leq \iint |q_1||\nabla_x K(q_1, q_2')| + \alpha|q_2||\nabla_y K(q_1', q_2)| \, \mu(dq')\mu(dq) \leq M,$$

where $M = \sup_{q_1 \in \mathcal{X}, q_2 \in \mathcal{Y}} |q_1||\nabla_x K(q_1, q_2')| + \alpha|q_2||\nabla_y K(q_1', q_2)|$. Also denoting by $R = \sup_{q \in \mathcal{X} \times \mathcal{Y}} |q|^2$, we see that for any $\alpha > 0$, $b(\cdot, \cdot)$ is $(\alpha, M + \alpha N)$-dissipative on average, as

$$\int \langle q, b(q,\mu) \rangle \, \mu(dq) + \alpha \int |q|^2 \, \mu(dq) \leq M + \alpha N.$$

**Optimistic algorithm fits Assumption 4**: Recall the iterates

$$q_{k+1}^i = q_k^i + \gamma_{k+1}\big(2b(q_k^i, \widehat{\mu}_k) - b(q_{k-1}^i, \widehat{\mu}_{k-1})\big) + \sqrt{2\gamma_{k+1}}\,\sigma \Xi_{k+1}^i,$$

where $\Xi_{k+1}^i = (\xi_{k+1}^i, \zeta_{k+1}^i)$. Notice that the bias of this iteration is

$$\varepsilon_{k+1}^i = b(q_k^i, \widehat{\mu}_k) - b(q_{k-1}^i, \widehat{\mu}_{k-1}).$$

For brevity, let us write $\mathscr{F}_k := \mathscr{F}_{\tau_k}$. We have

$$
\begin{aligned}
\mathbb{E}[|\varepsilon_{k+1}^i|^2 \mid \mathscr{F}_k] &= \mathbb{E}[|q_{k+1}^i - q_k^i - \gamma_{k+1} b(q_k^i, \widehat{\mu}_k) - \sqrt{2\gamma_{k+1}}\,\sigma \Xi_{k+1}^i|^2 \mid \mathscr{F}_k] \\
&\leq 3\,\mathbb{E}[|q_{k+1}^i - q_k^i|^2 \mid \mathscr{F}_k] + 3\gamma_{k+1}^2 \,\mathbb{E}[|b(q_k^i, \widehat{\mu}_k)|^2 \mid \mathscr{F}_k] + 6\gamma_{k+1}\tau(1 + \alpha)d.
\end{aligned}
$$

Moreover, we have

$$
\begin{aligned}
\mathbb{E}[|q_{k+1}^i - q_k^i|^2 \mid \mathscr{F}_k] &\leq 2\gamma_{k+1}^2 \,\mathbb{E}[|2b(q_k^i, \widehat{\mu}_k) - b(q_{k-1}^i, \widehat{\mu}_{k-1})|^2 \mid \mathscr{F}_k] + 2\gamma_{k+1}\tau(1 + \alpha)d \\
&= 2\gamma_{k+1}^2 \,\mathbb{E}[|b(q_k^i, \widehat{\mu}_k) + \varepsilon_{k+1}^i|^2 \mid \mathscr{F}_k] + 2\gamma_{k+1}\tau(1 + \alpha)d \\
&\leq 4\gamma_{k+1}^2 \,\mathbb{E}[|b(q_k^i, \widehat{\mu}_k)|^2 \mid \mathscr{F}_k] + 4\gamma_{k+1}^2 \,\mathbb{E}[|\varepsilon_{k+1}^i|^2 \mid \mathscr{F}_k] + 2\gamma_{k+1}\tau(1 + \alpha)d
\end{aligned}
$$

Combining the last two inequalities, we have

$$\mathbb{E}[|\varepsilon_{k+1}^i|^2 \mid \mathscr{F}_k] \le 3\Big(4\gamma_{k+1}^2 \,\mathbb{E}[|b(q_k^i,\widehat{\mu}_k)|^2 \mid \mathscr{F}_k] + 4\gamma_{k+1}^2\, \mathbb{E}[|\varepsilon_{k+1}^i|^2 \mid \mathscr{F}_k] + 2\gamma_{k+1}\tau(1+\alpha)d\Big)$$
$$+ 3\gamma_{k+1}^2 \,\mathbb{E}[|b(q_k^i,\widehat{\mu}_k)|^2 \mid \mathscr{F}_k] + 6\gamma_{k+1}\tau(1+\alpha)d$$
$$= 12\gamma_{k+1}^2 \,\mathbb{E}[|\varepsilon_{k+1}^i|^2 \mid \mathscr{F}_k] + 15\gamma_{k+1}^2 \,\mathbb{E}[|b(q_k^i,\widehat{\mu}_k)|^2 \mid \mathscr{F}_k] + 12\gamma_{k+1}\tau(1+\alpha)d$$

Since $\gamma_{k+1} \to 0$, we can assume that $12\gamma_{k+1}^2 \le 1/2$, which implies

$$\mathbb{E}[|\varepsilon_{k+1}^i|^2 \mid \mathscr{F}_k] \le 30\gamma_{k+1}^2 \,\mathbb{E}[|b(q_k^i,\widehat{\mu}_k)|^2 \mid \mathscr{F}_k] + 24\gamma_{k+1}\tau(1+\alpha)d,$$

which is exactly what we are after. ∎

## C.4 Kinetic Equations

*Proof of Corollary 4.* **Smoothness of the drift:** Let $x, y \in \mathbb{R}^d$ and $\mu, \nu \in \mathscr{P}_2(\mathbb{R}^d)$ and set $\pi$ be an optimal coupling between $\mu$ and $\nu$ (in $W_1$ sense). Then

$$|b(x,\mu) - b(y,\nu)| \le |\nabla V(x) - \nabla V(y)| + \left| \int \nabla W(x-z)\,\mu(dz) - \int \nabla W(y-z)\,\nu(dz) \right|.$$

By $L$-Lipschitzness of $\nabla V$, the first term is bounded by $L|x-y|$. For the second term, using the coupling, we can write it as

$$\left| \iint \nabla W(x-z_1) - \nabla W(y-z_2)\,\pi(dz_1, dz_2) \right| \le \iint |\nabla W(x-z_1) - \nabla W(y-z_2)|\,\pi(dz_1, dz_2)$$
$$\le L \iint |x-y+z_2-z_1|\,\pi(dz_1, dz_2)$$
$$\le L \iint |x-y| + |z_2-z_1|\,\pi(dz_1, dz_2)$$
$$\le L|x-y| + LW_1(\mu,\nu)$$
$$\le L|x-y| + LW_2(\mu,\nu).$$

Putting these together we get

$$|b(x,\mu) - b(y,\nu)| \le 2L(|x-y| + \mathcal{W}_2(\mu,\nu)).$$

**Average dissipativity of the drift:** First we show that for $x \in \mathbb{R}^d$ and a probability measure $\mu$, we have

$$\iint \langle x, \nabla W(x-y)\rangle\,\mu(dx)\mu(dy) \ge -M_W/2. \tag{C.1}$$

This holds, since

$$\iint \langle x, \nabla W(x-y)\rangle\,\mu(dx)\mu(dy)$$
$$= \iint \langle x-y+y, \nabla W(x-y)\rangle\,\mu(dx)\mu(dy)$$
$$= \iint \langle x-y, \nabla W(x-y)\rangle\,\mu(dx)\mu(dy) + \iint \langle y, \nabla W(x-y)\rangle\,\mu(dx)\mu(dy)$$
$$\ge -M_W + \iint \langle y, \nabla W(x-y)\rangle\,\mu(dx)\mu(dy)$$
$$\ge -M_W - \iint \langle x, \nabla W(x-y)\rangle\,\mu(dx)\mu(dy),$$

where in the penultimate inequality we used the assumption (which implies $\langle \nabla W(x), x\rangle \ge -M_W$), and in the last one, we used the that $W$ is symmetric (which implies $\nabla W(-z) = -\nabla W(z)$), and used Fubini's theorem to exchange integrals. Bringing the last term to the left and dividing by 2 shows (C.1).

To show average dissipativity, it suffices to observe

$$-\int \langle x, b(x, \mu) \rangle \, \mu(dx) = \int \langle x, \nabla V(x) \rangle \, \mu(dx) + \iint \langle x, \nabla W(x - y) \rangle \, \mu(dy)\mu(dx)$$

$$\geq \alpha \int |x|^2 \, \mu(dx) - \beta - M_W/2.$$

**Proximal algorithm fits Assumption 4**: Note that this implicit algorithm corresponds to the following proximal step

$$x_{k+1}^i = \arg\min_x \left\{ V(x) + \frac{1}{N} \sum_{j=1}^N W(x - x_k^j) + \frac{1}{2\gamma_{k+1}} \left| x - (x_k^i + \sqrt{2\gamma_{k+1}} \, \xi_{k+1}^i) \right|^2 \right\}.$$

By defining the perturbation as

$$P_{k+1}^i = \varepsilon_{k+1}^i = \nabla V(x_{k+1}^i) - \nabla V(x_k^i) + \frac{1}{N} \sum_{j=1}^N (\nabla W(x_{k+1}^i - x_k^j) - \nabla W(x_k^i - x_k^j)),$$

we see that the algorithm (Kin-Prox) fits the template (SAA). For brevity, let us write $\mathscr{F}_k := \mathscr{F}_{\tau_k}$. We only have to show that

$$\mathbb{E}[|\varepsilon_{k+1}|^2 \mid \mathscr{F}_k] = \sum_{i=1}^N \mathbb{E}[|\varepsilon_{k+1}^i|^2 \mid \mathscr{F}_k] = \mathcal{O}(\gamma_{k+1}^2 |b(x_k)|^2 + \gamma_{k+1}).$$

We have

$$\left| \varepsilon_{k+1}^i \right|^2 = \left| \nabla V(x_{k+1}^i) - \nabla V(x_k^i) + \frac{1}{N} \sum_{j=1}^N (\nabla W(x_{k+1}^i - x_k^j) - \nabla W(x_k^i - x_k^j)) \right|^2$$

$$\leq 2|\nabla V(x_{k+1}^i) - \nabla V(x_k^i)|^2 + \frac{2}{N} \sum_{j=1}^N \left| \nabla W(x_{k+1}^i - x_k^j) - \nabla W(x_k^i - x_k^j) \right|^2$$

$$\leq 2L^2 |x_{k+1}^i - x_k^i|^2 + \frac{2L^2}{N} \sum_{j=1}^N |x_{k+1}^i - x_k^i|^2$$

$$= 4L^2 |x_{k+1}^i - x_k^i|^2.$$

For brevity, let

$$f(x) = \nabla V(x) + \frac{1}{N} \sum_{j=1}^N \nabla W(x - x_k^j),$$

noticing that $\varepsilon_{k+1}^i = f(x_{k+1}^i) - f(x_k^i)$. By the update rule (Kin-Prox)

$$\mathbb{E}[|x_{k+1}^i - x_k^i|^2 \mid \mathscr{F}_k] \leq 2\gamma_{k+1}^2 \mathbb{E}[|f(x_{k+1}^i)|^2 \mid \mathscr{F}_k] + 4\gamma_{k+1}d.$$

Moreover, we have that $|f(x_{k+1}^i)|^2 \leq 2|f(x_{k+1}^i) - f(x_k^i)|^2 + 2|f(x_k^i)|^2$. Since $\gamma_{k+1} \to 0$, we can assume that $16L^2\gamma_{k+1}^2 < \frac{1}{2}$. All in all, this gives

$$\mathbb{E}[|\varepsilon_{k+1}^i|^2 \mid \mathscr{F}_k] \leq 4L^2 \mathbb{E}[|x_{k+1}^i - x_k^i|^2 \mid \mathscr{F}_k]$$

$$\leq 8L^2\gamma_{k+1}^2 \mathbb{E}[|f(x_{k+1}^i)|^2 \mid \mathscr{F}_k] + 16L^2\gamma_{k+1}d$$

$$\leq 16L^2\gamma_{k+1}^2 \mathbb{E}[|f(x_{k+1}^i) - f(x_k^i)|^2 \mid \mathscr{F}_k] + 16L^2\gamma_{k+1}^2 |f(x_k^i)|^2 + 16L^2\gamma_{k+1}d$$

$$\leq 16L^2\gamma_{k+1}^2 \mathbb{E}[|\varepsilon_{k+1}^i|^2 \mid \mathscr{F}_k] + 16L^2\gamma_{k+1}^2 |f(x_k^i)|^2 + 16L^2\gamma_{k+1}d$$

$$\leq \frac{1}{2} \mathbb{E}[|\varepsilon_{k+1}^i|^2 \mid \mathscr{F}_k] + 16L^2\gamma_{k+1}^2 |f(x_k^i)|^2 + 16L^2\gamma_{k+1}d.$$

This implies that

$$\mathbb{E}[|\varepsilon_{k+1}^i|^2 \mid \mathscr{F}_k] \leq 32L^2\gamma_{k+1}^2 |f(x_k^i)|^2 + 32L^2\gamma_{k+1}d.$$

Summing over $i$ and observing that $\sum |f(x_k^i)|^2 = |b(x_k)|^2$ concludes the proof. ∎

