# OpenReview forum: "Stochastic Approximation Algorithms for Systems of Interacting Particles"
_NeurIPS.cc/2023/Conference — NeurIPS 2023 poster_

### Official Review · Reviewer_77LN · 2023-07-02

**Soundness:** 3 good
**Presentation:** 3 good
**Contribution:** 2 fair
**Rating:** 6
**Confidence:** 4

**Summary:**

This paper analyses discretisations of mean-field type SDEs arising in several areas of machine learning. The main contribution is a convergence result (Theorem 1) stating that under appropriate conditions on the drift and diffusion coefficients, the discretised dynamics convergence in 2-Wasserstein distance, in an infinite time horizon limit, to the continuous-time system of interacting particles. Using the classical uniform propagation of chaos for interacting particle systems to show convergence to the mean field dynamics. Applications to Two-Layer NNs training by SGD, Stein Variational Gradient Descent, Two-Player Zero-sum Continuous Games and Kinetic Equations are presented.

**Strengths:**

As outlined by the authors, SDEs of mean-field type arise in several areas of machine learning and statistics as continuous-time and infinite-number-of-particle limits of discrete stochastic difference equations. Therefore, a convergence analysis of the discrete schemes to the their continuous-time counterparts is of high relevance and importance within the modern machine learning landscape. The paper is well-written and clear, its structure is easy to follow. The four examples on Two-Layer NNs, Stein Variational Gradient Descent, Two-Player Zero-sum Continuous Games and on Kinetic Equations nicely demonstrate the applicability of the main result (Theorem 1) to several topics/areas of modern machine learning.

**Weaknesses:**

There is a rich body of literature on SDE discretisation scheme for McKean-Vlasov SDEs and interacting particle systems [1, 2, 3], that is completely ignored by the authors. The results of these papers concern convergence of Euler-Marayama and/or Milstein type numerical schemes to the limiting mean-field equations when the step size of the solver goes to zero and the number of particles goes to infinity. As far as I see, the main difference in the anlyses is in the notion of convergence in time: the authors consider as convergence criterion the *Wasserstein asymptotic pseudotrajectory* (WAPT), which is a large time behaviour from dynamical systems theory, while in the aforementioned series of paper the convergence is in terms of discretisation step, which is more classical in (numerical) stochastic analysis. Albeit the two notions of convergences are different, I think an in-depth discussion and comparison between the two is required.

I invite the authors to initiate a conversation on the above during the rebuttal period. If a rigorous and fair comparison/discussion is eventually presented, I will happily increase my rating.

**References**

[1] Bao, Jianhai, et al. "First-order convergence of Milstein schemes for McKean–Vlasov equations and interacting particle systems." Proceedings of the Royal Society A 477.2245 (2021): 20200258.

[2] Reisinger, Christoph, and Wolfgang Stockinger. "An adaptive Euler–Maruyama scheme for McKean–Vlasov SDEs with super-linear growth and application to the mean-field FitzHugh–Nagumo model." Journal of Computational and Applied Mathematics 400 (2022): 113725.

[3] Leobacher, Gunther, Christoph Reisinger, and Wolfgang Stockinger. "Well-posedness and numerical schemes for one-dimensional McKean–Vlasov equations and interacting particle systems with discontinuous drift." BIT Numerical Mathematics 62.4 (2022): 1505-1549.

**Questions:**

- Can the authors comment on the choice of 2-Wasserstein distance in the WAPT convergence criterion?
- The authors make repetitive use of the notion of a *limit set*; albeit this is classical in the theory of dynamical systems, it is worth defining it in the main body of the paper.
- The paper would benefit from the addition of an *idea-of-proof* paragraph following Theorem 1 (still defering the details to the appendix).
- (This is not necessarily a question targeted to this paper, but more generally to the community working at the intersection between NNs and mean-field dynamics) What are the issues in analysing more-than-two layers feedforward NNs using the language of mean-field SDEs? What about more general architectures such ResNets, RNNs etc. ?

**Limitations:**

The authors adequately addressed limitations of their contribution and discussed future work.

---

> ### Author Rebuttal · Authors · 2023-08-07
>
> Thank you for your valuable and insightful review, especially for pointing out the missing references. We will incorporate Reviewer's recommendation in the revision accordingly.
>
> > There is a rich body of literature on SDE discretisation scheme for McKean-Vlasov SDEs and interacting particle systems [1, 2, 3], that is completely ignored by the authors. The results of these papers concern convergence of Euler-Marayama and/or Milstein type numerical schemes to the limiting mean-field equations when the step size of the solver goes to zero and the number of particles goes to infinity. As far as I see, the main difference in the anlyses is in the notion of convergence in time: the authors consider as convergence criterion the Wasserstein asymptotic pseudotrajectory (WAPT), which is a large time behaviour from dynamical systems theory, while in the aforementioned series of paper the convergence is in terms of discretisation step, which is more classical in (numerical) stochastic analysis. Albeit the two notions of convergences are different, I think an in-depth discussion and comparison between the two is required.
>
> We sincerely appreciate the Reviewer for bringing these references to our attention, which we had indeed overlooked. It is important to clarify several significant differences between our work and the references [1, 2, 3]:
>
>
> - **Generic Stochastic and Biased Drift Oracles**: Our primary focus lies in generic stochastic and biased drift oracles denoted by $b(\cdot,\cdot)$, while [1, 2, 3] concentrate on deterministic and unbiased drift oracles. Consequently, our algorithmic framework is substantially more general than theirs.
>
> - **Asymptotic vs Finite-Time Bounds**: On the other hand, our convergence results provide asymptotic guarantees, whereas the results in [1, 2, 3] offer much stronger bounds, explicitly controlling the $\mathcal{W}_2$ error in finite time.
>
> - **Incomparable Assumptions**: Apart from the aforementioned differences, our work and [1, 2, 3] rely on incomparable assumptions. For instance, we impose global Lipschitz drifts, whereas [1, 2, 3] can handle more general drifts with only one-sided Lipschitzness. On the other hand, our milder growth condition in Assumption 2 requires control on average, while the stronger pointwise controls are assumed in [1, 2, 3].
>
> In light of these distinctions, we believe that our paper complements the references pointed out by the Reviewer. Moreover, these works raise an intriguing research question: Can the Milstein schemes be adapted as stochastic approximation schemes within our framework, potentially leading to stochastic versions of Milstein schemes that enhance computational efficiency? We look forward to exploring this avenue for future research.
>
> > Can the authors comment on the choice of 2-Wasserstein distance in the WAPT convergence criterion?
>
> There are two major reasons for adopting the Wasserstein distances in our framework:
>
> - It is a popular metric in the propagation of chaos literature, which our framework relies on. Adopting the Wasserstein metrics therefore allows for a seamless transition from stochastic approximation schemes (finite step-size + finite particles) to its mean-field continuous-time limits (infinitesimal step-size + infinite particles) by combining our theory and the propagation of chaos results.
>
> - It is important that 2-Wasserstein space is a **metric space** on which McKean–Vlasov equations can be seen as a **flow**, both aspects indispensable for invoking the dynamical system theory of Benaïm and Hirsch.
>
> > The authors make repetitive use of the notion of a limit set; albeit this is classical in the theory of dynamical systems, it is worth defining it in the main body of the paper.
>
> We agree on this point and we intend to include relevant definitions in the revision.
>
> > The paper would benefit from the addition of an idea-of-proof paragraph following Theorem 1 (still defering the details to the appendix).
>
> We agree with the Reviewer. The main issue preventing us from this was the page limit, which can be easily addressed given an extra page on the camera-ready version.
>
> > (This is not necessarily a question targeted to this paper, but more generally to the community working at the intersection between NNs and mean-field dynamics) What are the issues in analysing more-than-two layers feedforward NNs using the language of mean-field SDEs? What about more general architectures such ResNets, RNNs etc.?
>
> It is possible to extend our framework beyond two layers: The mean-field limit for multilayer or structured neural networks is a research area with several notable contributions. In particular, [NP], [SS], and [AOY] have explored the mean-field limit for deep networks, while [F] has addressed this topic in the context of ResNets.
>
> The rationale behind our selection of a 2-layer neural network lies in the pursuit of clarity in representation. By focusing on this simpler architecture, we aim to provide a more straightforward and accessible presentation of our work.
>
> ---
> We hope that the above addresses your questions - but please let us know if any of the above is not sufficiently clear.
>
> Thank you again for your input and positive evaluation,
>
> The authors
>
>
> **References:**
>
> [NP] A Rigorous Framework for the Mean Field Limit of Multilayer Neural Networks by Phan-Minh Nguyen, Huy Tuan Pham.
>
> [SS] Mean Field Analysis of Deep Neural Networks by Justin Sirignano and Konstantinos Spiliopoulos.
>
> [AOY] A mean-field limit for certain deep neural networks by Dyego Araújo, Roberto I. Oliveira, and Daniel Yukimura.
>
> [F] Modeling from Features: a Mean-field Framework for Over-parameterized Deep Neural Networks by Fang et al.

---

> > ### Comment · Reviewer_77LN · 2023-08-12
> >
> > I thank the authors for their responses which have addressed my concernes. I do not have any further questions. I keep my rating unchanged, and I'm considering raising it to 7. I will make a final decision after consultation with other reviewers and AC.

---

> > > ### Author Response · Authors · 2023-08-20
> > > **Official Comment by Authors**
> > >
> > > We extend our gratitude to the Reviewer for pointing out the missing references and for the thoughtful consideration of a potential score increase. We will integrate these discussions into our forthcoming revision.

---

### Official Review · Reviewer_S8Cs · 2023-07-06

**Soundness:** 4 excellent
**Presentation:** 4 excellent
**Contribution:** 2 fair
**Rating:** 6
**Confidence:** 2

**Summary:**

This paper develops a theoretical mathematical framework to characterize the convergence properties of discrete particle systems to their mean-field limit.

**Strengths:**

The mathematical theory in this paper is beyond my scope, but it appears to be mathematically sound. The paper is well-written.

**Weaknesses:**

I believe that this paper would benefit from some applied tests/results to show practical relevance. For example, how would it help training a GAN? Do the theoretical convergence results help actual training? How do the results compare to actual training? Are the bounds tight relative to actual convergence?

**Questions:**

I think it would be helpful to provide some examples of how these these theoretical results can help inform and guide ML development, etc. For example, how can I use these convergence guarantees as a design a NN?

**Limitations:**

Limitations do not seem to be explicitly addressed by the authors.

---

> ### Author Rebuttal · Authors · 2023-08-07
>
> We thank the Reviewer for raising issue of practical relevance. We have taken this concern seriously and made the necessary adjustments to address it in the rebuttal below, which will be incorporated into the revision.
>
> Having addressed the concerns and made the appropriate changes, we sincerely hope for a re-evaluation of our work based on the constructive discussions we have engaged in. We are open and eager for any further discussions or feedback that can contribute to the improvement and practical applicability of our submission.
>
> > I believe that this paper would benefit from some applied tests/results to show practical relevance. For example, how would it help training a GAN? Do the theoretical convergence results help actual training? How do the results compare to actual training? Are the bounds tight relative to actual convergence?
>
> >Questions:
> I think it would be helpful to provide some examples of how these these theoretical results can help inform and guide ML development, etc. For example, how can I use these convergence guarantees as a design a NN?
>
>
> We address these concerns together. The strength of our frameworks lies not in the design of neural networks but rather in their **algorithmic flexibility**. We provide two justifications to support this claim:
>
> 1. **Providing rigorous guarantees for existing interacting particle systems:** In the machine learning community, exploiting stochastic gradients is a common practice for training large-scale neural networks, even when the algorithm's motivation and analysis are based on *deterministic* gradients. This is evident in methods like SVGD, where literature on stochastic gradients is scarce (cf. Section 4.2). In this context, our framework establishes rigorous convergence guarantees for the important stochastic SVGD methods under the mild assumption that the noise has a finite variance, ensuring the applicability of these popular schemes.
>
> 2. **Enabling algorithmic design:** We present a pertinent example in multi-agent learning that is not included in the current version of our submission. The (GDA$\_k$) and (OGDA$\_k$) schemes in our paper rely on *simultaneous* updates, i.e., $(x_k, y_k) \rightarrow (x_{k+1}, y_{k+1})$, as dictated by existing theory. However, empirical evidence suggests that *alternating* updates $(x_k, y_k) \rightarrow (x_{k}, y_{k+1}) \rightarrow (x_{k+1}, y_{k+1})$ often performs better. Our framework allows for this flexibility, as it is easy to cast alternating (GDA$\_k$) and (OGDA$\_k$) stochastic approximation schemes satisfying **Assumption 4**, and thus, convergence is guaranteed according to our theory.
>
>
>     In a broader context, our algorithmic template facilitates flexible design by merely verifying a few straightforward assumptions. This empowers researchers and practitioners to explore and develop novel algorithms that suit specific requirements and scenarios, such as in the game-theoretic settings mentioned earlier
>
>
> > Limitations do not seem to be explicitly addressed by the authors.
>
> We will incorporate the above discussions into our forthcoming revision and make sure to highlight the limitations of our framework.
>
> ---
> We hope that the above addresses your questions - but please let us know if any of the above is not sufficiently clear.
>
> Thank you again for your input and constructive criticism,
>
> The authors

---

> > ### Comment · Reviewer_S8Cs · 2023-08-18
> >
> > Thank you for the clarification. I will raise my rating to a 6.

---

> > > ### Author Response · Authors · 2023-08-20
> > > **Official Comment by Authors**
> > >
> > > We appreciate your initiative in highlighting the practicality concerns within our theory. Furthermore, we extend our gratitude for your favorable re-assessment.
> > >
> > > We kindly ask the Reviewer's attention to the pending score upgrade that was promised. Your assistance in fulfilling this commitment would be greatly appreciated.

---

> > > > ### Comment · Reviewer_S8Cs · 2023-08-21
> > > >
> > > > I have raised my rating to 6.

---

> > > > > ### Author Response · Authors · 2023-08-21
> > > > > **Official Comment by Authors**
> > > > >
> > > > > Thank you for the prompt response.

---

### Official Review · Reviewer_UKxL · 2023-07-07

**Soundness:** 3 good
**Presentation:** 3 good
**Contribution:** 2 fair
**Rating:** 6
**Confidence:** 4

**Summary:**

This work considers the convergence of discrete time interacting particle systems to their respective continuous time limits (i.e, McKean-Vlasov type equations) under general assumptions which are applicable to varied contexts like neural networks, kinetic theory, game theory and sampling algorithms.  The finite particle + finite step size algorithms are considered to be stochastic approximations of the mean field limit and convergence is analyzed in terms of dynamical systems theory.  This work considers the notion of weak asymptotic pseudo-trajectory to show that the stochastic approximations are close to mean field limit under general conditions. These are then applied to various specific contexts like neural networks and SVGD to derive convergence bounds.

**Strengths:**

The generality of the assumptions and the framework is the main contribution of this work. This enables the authors to derive several useful results in varied domains under a common framework. This can be a useful tool in establishing convergence to mean field limits in new problems without requiring elementary analysis.

**Weaknesses:**

1. The notation in the algorithmic template is extremely confusing. Shortening $b(x,\mu)$ to just $b(x)$ makes it very confusing. Population level SA is also a bad terminology since it confuses the reader about whether this is the mean-field limit (i.e, $n \to \infty$) or not (because the mean field limit is often referred to as the population limit).

2. WAPT as the notion of convergence requires more justification. This is so since the continuous process begins at $X_t$, the $t$-th time instant of the discrete time process and the uniform convergence is established as $t \to \infty$. What if the initial deviation in the stochastic approximation ensures that $X_t$ itself is not likely to be reached by PSDE ?

3. Assumption 5 is a bit non-standard. Also, I think there is a typo here. Assumption (11) is not satisfied for any decreasing step size sequence since $\gamma_{k+1}/\gamma_{k+2} > 1$. Please clarify and state what exact step sizes are allowed.

4. Under specific settings, much stronger results can be derived for convergence when the algorithm is designed specially or under specific assumptions like logarithmic sobolev inequalities (even with finite particles and constant step sizes). This framework precludes such analyses.  (See [A1,A2]). I am not very well versed in the game theory or kinetic theory literature, so I will abstain from commenting on these results.

[A1]  Convergence of mean-field Langevin dynamics: Time and space discretization, stochastic gradient, and variance reduction.

[A2]  Provably Fast Finite Particle Variants of SVGD via Virtual Particle Stochastic Approximation


**Questions:**

Please answer the questions posed in the weaknesses section.

**Limitations:**

The authors have a good discussion on the applicability of their work. I think they should discuss the drawbacks of WAPT better.

---

> ### Author Rebuttal · Authors · 2023-08-07
>
> Thank you for your valuable input and remarks. We are dedicated to addressing your concerns through revisions in our upcoming review. Having taken all your feedback into account, we kindly ask for your consideration in potentially revising the score.
>
> > The notation in the algorithmic template is extremely confusing. Shortening b(x,mu) to just b(x) makes it very confusing. Population level SA is also a bad terminology since it confuses the reader about whether this is the mean-field limit (i.e, ) or not (because the mean field limit is often referred to as the population limit).
>
> We thank the reviewer for this comment. We can use "aggregate drift/diffusion" instead of "population level drift/diffusion" and also make the notation for the aggregated drift and diffusion boldface to avoid further confusion.
>
> > WAPT as the notion of convergence requires more justification. This is so since the continuous process begins at , the -th time instant of the discrete time process and the uniform convergence is established as . What if the initial deviation in the stochastic approximation ensures that itself is not likely to be reached by PSDE ? I think they should discuss the drawbacks of WAPT better.
>
> We thank the Reviewer for bringing forth this question, leading us to recognize that our presentation can be substantially improved by clarifying this misunderstanding: WAPT is **not** intended to serve as a notion of convergence itself. Rather, we **prove** that popular schemes in practice **are** WAPTs, which **implies** convergence to the desirable measures in the standard 2-Wasserstein metric.
>
> To conclude, the notion of WAPT is an important intermediate step in our analysis framework, not the final convergence result. This nuanced perspective will be highlighted in our forthcoming revision.
>
> > Assumption 5 is a bit non-standard. Also, I think there is a typo here. Assumption (11) is not satisfied for any decreasing step size sequence since... Please clarify and state what exact step sizes are allowed.
>
> Thanks a lot for pointing out this typo: Equation (11) should read $\gamma_{k+1}/\gamma_k + P \gamma_k\gamma_{k+1} < 1 - \gamma_k$. We will fix this in the final version.
>
> We also remark that this assumption is not restrictive. For example, one can show that it is met by step-sizes as slow as $1 / (\sqrt{k} \log k)$.
>
> > Under specific settings, much stronger results can be derived for convergence when the algorithm is designed specially or under specific assumptions like logarithmic sobolev inequalities (even with finite particles and constant step sizes). This framework precludes such analyses. (See [A1,A2]). I am not very well versed in the game theory or kinetic theory literature, so I will abstain from commenting on these results.
> [A1] Convergence of mean-field Langevin dynamics: Time and space discretization, stochastic gradient, and variance reduction.
> [A2] Provably Fast Finite Particle Variants of SVGD via Virtual Particle Stochastic Approximation
>
> Thank you for your insightful comments and references. While the Reviewer has rightfully pointed out that much stronger results can be derived under additional assumptions, we argue that our analysis *complements* instead of "precluding" stronger assumptions such as LSI: In scenarios where establishing LSIs proves challenging, such as in multi-agent systems or SVGD, our theory demonstrates that existing schemes still converge under remarkably mild conditions.
>
> Additionally, we highlight that when strong assumptions like LSIs are present, it is possible to enhance the notion of asymptotic pseudo-trajectories to its non-asymptotic counterpart, known as the $\lambda$-pseudo-trajectories. However, as our primary focus lies in the generic setting where such assumptions are unavailable, we have chosen to defer these studies to future work.
>
> ---
> We hope that the above addresses your questions - but please let us know if any of the above is not sufficiently clear.
>
> Thank you again for your input and positive evaluation,
>
> The authors

---

> > ### Comment · Reviewer_UKxL · 2023-08-18
> > **Thank you**
> >
> > Thanks for the response. I am satisfied with the rebuttal and raise my score to a 6.

---

> > > ### Author Response · Authors · 2023-08-20
> > > **Official Comment by Authors**
> > >
> > > Thank you for bringing several issues in our presentation to our attention. We also value your updated assessment.

---

### Official Review · Reviewer_MJDT · 2023-07-13

**Soundness:** 4 excellent
**Presentation:** 4 excellent
**Contribution:** 4 excellent
**Rating:** 9
**Confidence:** 3

**Summary:**

This paper fills a theoretical gap between the application of ideas interacting particle systems to algorithms in machine learning--algorithms, like SGVD, that are almost always realized as discrete-time routines with a finite number of particles--and the substantial existing body of theoretical work on finite particle systems with continuous dynamics. These latter works have yielded valuable insights about, e.g., the training process of two-layer neural networks, algorithm design for approximate Bayesian inference, or the nature of equilibria in games. However, they have not rigorously established the convergence of discrete-time to continuous. This paper establishes that convergence via Benaïm and Hirsch's notion of a Wasserstein asymptotic pseudo-trajectory (WAPT), which gives a measure of asymptotic closeness (in the Wasserstein-2 sense) between two stochastic processes. Specifically, via Theorem 1, the convergence of the family of (discrete-time) stochastic approximation algorithms (SAA) is reduced to its continuous time counterpart. Combining Theorem 1 with existing results in the literature yields the conclusion that the empirical distribution of particles following the discrete-time SAA converges to the mean-field solution, as desired.

**Strengths:**

The central originality of this paper lies in its adaptation of WAPT to solve an open problem in the mean-field theory of discrete-time IPS in machine learning.
Broadly, I found the clarity and elegance of the mathematical exposition to be exceptionally good. The paper persuasively argues for the importance of a rigorous theory of convergence, and smoothly introduces concepts and definitions needed for understanding Theorem 1, while re-orienting the reader by summarizing previous results at effective moments. After stating the Theorem, the applications of the theory to two-layer NNs, SGVD, games, and kinetic equations were clear and enlightening.
The significance of the presented comprehensive framework is high, as the future directions section makes clear.

**Weaknesses:**

I found the paper without major weaknesses. In terms of the overall presentation of results, I was surprised to see interacting particle systems (IPS) as the frame for this theory rather than simply continuous-time Markov jump processes. I can see the value in specializing to interacting particle systems, but some readers may be deeply acquainted with continuous Markov jump processes and be largely unaware of the IPS literature. Bringing that connection onto the screen by mentioning it in the technical background may help orient readers with a more general stochastic process background.

In the same vein, a brief mention of the relationship to multi-agent systems could be of value in ensuring that this work reaches the wide readership for which its theory is relevant.

**Questions:**

- For readers unfamiliar with pseudo-trajectories from the dynamical systems literature, a brief footnote or aside giving a terse formal definition may be helpful. The intuitive explanation is mostly clear--the requirement than the orbit X(t) closely tracks the flow over arbitrarily long time intervals T with arbitrary precision--but a formal statement that uses a constant like \lambda (as I see in line 352!) would likely not strain the reader too much.
- It may be valuable, in the future directions section or in an appendix, to discuss what kinds of algorithms do not fall under the algorithmic template of the SAA family, as a way to highlight other gaps in the existing mean-field theory.

**Limitations:**

The authors have adequately address the limitations of their work. I see no potential negative societal impacts.

---

> ### Author Rebuttal · Authors · 2023-08-07
>
> Thank you for your input and remarks. We reply to your questions below, and we will revise our manuscript accordingly in the upcoming revision.
>
> > In terms of the overall presentation of results, I was surprised to see interacting particle systems (IPS) as the frame for this theory rather than simply continuous-time Markov jump processes. I can see the value in specializing to interacting particle systems, but some readers may be deeply acquainted with continuous Markov jump processes and be largely unaware of the IPS literature. Bringing that connection onto the screen by mentioning it in the technical background may help orient readers with a more general stochastic process background. In the same vein, a brief mention of the relationship to multi-agent systems could be of value in ensuring that this work reaches the wide readership for which its theory is relevant.
>
> We thank the reviewer for bringing this up. Indeed, we have altogether omitted the discussion related to Markov jump processes via passing to the mean-field limit $N\rightarrow\infty$. A similar remark holds for the multi-agent perspective: The mean-field game perspective allows us to bypass it analytically, but we agree that expanding upon these perspectives could enhance the clarity and understanding of our work.
>
> > For readers unfamiliar with pseudo-trajectories from the dynamical systems literature, a brief footnote or aside giving a terse formal definition may be helpful. The intuitive explanation is mostly clear--the requirement than the orbit X(t) closely tracks the flow over arbitrarily long time intervals T with arbitrary precision--but a formal statement that uses a constant like $\lambda$ (as I see in line 352!) would likely not strain the reader too much.
>
> We thank the Reviewer for this reminder: We have indeed defined the concept of a WAPT early-on (line 139). The other concept ($\lambda$-pseudotrajectory) is stronger than the usual APT, and gives stronger results such as asymptotic rates, which is left for future work (see lines 351-354).
>
> > It may be valuable, in the future directions section or in an appendix, to discuss what kinds of algorithms do not fall under the algorithmic template of the SAA family, as a way to highlight other gaps in the existing mean-field theory.
>
> A prime illustration of this is the *adaptive* schemes, like Adam, for training wide two-layer neural networks. Yet another very important example that our theory does not have guarantees for is the Ensemble Kalman Sampler, which we mentioned in the conclusion. Even though this algorithm follows the SAA template, one does not know *a priori* if the diffusion coefficient is bounded in Hilbert-Schmidt norm. Proving convergence of these algorithms is challenging (even in continuous-time) and we do not yet know how to deal with such problems.
>
> ---
> We hope that the above addresses your questions - but please let us know if any of the above is not sufficiently clear.
>
> Thank you again for your input and positive evaluation,
>
> The authors

---

### Author Rebuttal · Authors · 2023-08-09

Dear AC, dear reviewers,

We deeply appreciate your time, input, and thoughtful critiques, as well as your positive evaluation. Your contributions have our sincere gratitude, and all your questions are addressed in a separate point-by-point thread below.

A focal concern that has emerged pertains to the practical relevance of our paper. In this context, we wish to emphasize that our work brings forth two pivotal contributions with direct relevance for practitioners:

1. **Ensuring guarantees for popular heuristics:** Numerous widely-employed interacting particle systems, such as several stochastic variants of SVGD, currently lack convergence guarantees. Our paper fills this gap by establishing their rigorous convergence, thereby solidifying the reliability of these approaches.

2. **Devising novel schemes:** Through a simple validation of **Assumption 4**, our framework furnishes a template for inspiring novel schemes. An illustration of this potential is elucidated in the discussion with Reviewer S8Cs, where we delve into a concrete application within game theory.

For a comprehensive exploration of the remaining points, we refer you to the specific threads tailored to each reviewer's concerns. As we proceed to the discussion phase, we eagerly anticipate any further inquiries that may arise.

With the utmost appreciation,

The authors

---

### Decision · Program_Chairs · 2023-09-21

**Decision:**

Accept (poster)

**Comment:**

This paper derives an asymptotic convergence analysis of mean field Langevin dynamics. More precisely, it is shown that the discrete time dynamics with finite particles converges that with an infinitely many particles (mean field limit) as the number of particles goes to infinity, and the convergence is analyzed in the infinite time horizon limit (Wasserstein asymptotic pseudotrajectory).
The authors showed the result under a rather milder assumptions than the usual log-Sobolev inequality, which makes difference from the recent development on the uniform-in-time propagation of chaos. It gives another piece of convergence guarantee on the interacting particle systems. I would like to recommend acceptance.